# Convergence Rates of Stochastic Gradient Descent under Infinite Noise Variance

**Hongjian Wang**[*]
Carnegie Mellon University
hjnwang@cmu.edu

**Mert Gürbüzbalaban**
Rutgers University
mg1366@rutgers.edu

**Lingjiong Zhu**
Florida State University
zhu@math.fsu.edu

**Umut Şimşekli**
INRIA & ENS – PSL Research University
umut.simsekli@inria.fr

**Murat A. Erdogdu**
University of Toronto & Vector Institute
erdogdu@cs.toronto.edu

## Abstract

Recent studies have provided both empirical and theoretical evidence illustrating that heavy tails can emerge in stochastic gradient descent (SGD) in various scenarios. Such heavy tails potentially result in iterates with diverging variance, which hinders the use of conventional convergence analysis techniques that rely on the existence of the second-order moments. In this paper, we provide convergence guarantees for SGD under a state-dependent and heavy-tailed noise with a potentially infinite variance, for a class of strongly convex objectives. In the case where the $p$-th moment of the noise exists for some $p \in [1, 2)$, we first identify a condition on the Hessian, coined '$p$-positive (semi-)definiteness', that leads to an interesting interpolation between the positive semi-definite cone ($p = 2$) and the cone of diagonally dominant matrices with non-negative diagonal entries ($p = 1$). Under this condition, we provide a convergence rate for the distance to the global optimum in $L^p$. Furthermore, we provide a generalized central limit theorem, which shows that the properly scaled Polyak-Ruppert averaging converges weakly to a multivariate $\alpha$-stable random vector. Our results indicate that even under heavy-tailed noise with infinite variance, SGD can converge to the global optimum without necessitating any modification neither to the loss function nor to the algorithm itself, as typically required in robust statistics. We demonstrate the implications of our results over misspecified models, in the presence of heavy-tailed data.

## 1 Introduction

We consider the unconstrained minimization problem

$$\underset{\boldsymbol{x} \in \mathbb{R}^n}{\text{minimize}} \; f(\boldsymbol{x}), \tag{1.1}$$

using the stochastic gradient descent (SGD) algorithm. Initialized at $\boldsymbol{x}_0 \in \mathbb{R}^n$, the SGD algorithm is given by the iterations,

$$\boldsymbol{x}_{t+1} = \boldsymbol{x}_t - \gamma_{t+1}\big(\boldsymbol{\nabla} f(\boldsymbol{x}_t) + \boldsymbol{\xi}_{t+1}(\boldsymbol{x}_t)\big), \;\; t = 0, 1, 2, ... \tag{1.2}$$

where $\{\gamma_t\}_{t \in \mathbb{N}^+}$ denotes the step-size sequence, and $\{\boldsymbol{\xi}_t\}_{t \in \mathbb{N}^+}$ is a martingale difference sequence adapted to a filtration $\{\mathcal{F}_t\}_{t \in \mathbb{N}}$, characterizing the noise in the gradient (the sequence $\{\boldsymbol{x}_t\}_{t \in \mathbb{N}}$ is also adapted to the same filtration, if we assume $\boldsymbol{x}_0$ is $\mathcal{F}_0$-measurable). Our focus is on the case where the noise is state dependent, and its variance is infinite, i.e., $\mathbb{E}\big[\|\boldsymbol{\xi}_t\|_2^2\big] = \infty$.

---

[*]Work partially conducted while affiliated with the Vector Institute.

35th Conference on Neural Information Processing Systems (NeurIPS 2021).

Many problems in modern statistical learning can be written in the form (1.1), where $f(\boldsymbol{x})$ typically corresponds to the population risk, that is, $f(\boldsymbol{x}) := \mathbb{E}_{z \sim \nu}[\ell(\boldsymbol{x}, z)]$ for a given loss function $\ell$ and an unknown data distribution $\nu$. In practice, one observes independent and identically distributed (i.i.d.) samples $z_i \sim \nu$ for $i \in [n]$, and estimates the population gradient $\boldsymbol{\nabla} f(\boldsymbol{x})$ with a noisy gradient at each iteration, which is based on an empirical average over a subset of the samples $\{z_i\}_{i \in [n]}$. Due to its simplicity, superior generalization performance, and well-understood theoretical guarantees, SGD has been the method of choice for minimization problems arising in statistical machine learning.

Starting from the pioneering works of Robbins and Monro [1951], Chung [1954], Sacks [1958], Fabian [1968], Ruppert [1988], Shapiro [1989], Polyak and Juditsky [1992], theoretical properties of the SGD algorithm and its variants have been receiving a growing attention under different scenarios. Recent works, for example Tripuraneni et al. [2018], Su and Zhu [2018], Duchi and Ruan [2021], Toulis and Airoldi [2017], Fang et al. [2018], Anastasiou et al. [2019], Yu et al. [2020] established convergence rates for SGD in various settings. By building on the analysis of Polyak and Juditsky [1992] to prove a central limit theorem (CLT) for the Polyak-Ruppert averaging, these works led to novel methodologies to compute confidence intervals using SGD. However, a recurring assumption in this line of work is the finite noise variance, which may be violated frequently in modern frameworks.

Heavy-tailed behavior in statistical methodology may naturally arise from the underlying model, or through the iterative optimization algorithm used during model training. In robust statistics, one often encounters heavy-tailed noise behavior in data, which in conjunction with standard loss functions leads to infinite noise variance in SGD. Very recently, heavy-tailed behavior is shown to emerge from the multiplicative noise in SGD, when the step-size is large and/or the batch-size is small [Hodgkinson and Mahoney, 2021, Gürbüzbalaban et al., 2021]. On the other hand, there is strong empirical evidence in modern machine learning that the gradient noise often exhibits a heavy-tailed behavior, which indicates an infinite variance. For example, this is observed in fully connected and convolutional neural networks [Şimşekli et al., 2019, Gürbüzbalaban and Hu, 2021] as well as recurrent neural networks [Zhang et al., 2020]. Thus, understanding the behavior of SGD under infinite noise variance becomes extremely important for at least two reasons. A *computational complexity reason:* modern machine learning and robust statistics frameworks lead to heavy-tailed behavior in SGD; thus, understanding the performance of this algorithm in terms of precise convergence rates as well as the required conditions on the step-size sequence as a function of the 'heaviness' of the tail become crucial in this setup. A *statistical reason:* many inference methods that rely on Polyak-Ruppert averaging utilize a CLT which holds under finite noise variance (see e.g. online bootstrap and variance estimation approaches [Fang et al., 2018, Su and Zhu, 2018, Chen et al., 2020]). Using the same methodology in the aforementioned modern frameworks (under heavy-tailed noise) will ultimately result in incorrect confidence intervals, jeopardizing the statistical procedure. Thus, establishing the limit distribution in this setting is of great importance.

In this work, we study the behavior of the SGD algorithm with diminishing step-sizes for a class of strongly convex problems when the noise variance is infinite. We establish the convergence rates of the SGD iterates towards the global minimum, and identify a sufficient condition on the Hessian of $f$, which interpolates between the positive semi-definite cone and the cone of diagonally dominant matrices (with non-negative diagonal entries). We further study the Polyak-Ruppert averaging of the SGD iterates, and show that the limit distribution is a multivariate $\alpha$-stable distribution. We illustrate our theory on linear regression and generalized linear models, demonstrating how to verify the conditions of our theorems. Perhaps surprisingly, our results show that even under heavy-tailed noise with infinite variance, SGD with diminishing step-sizes can converge to the global optimum without requiring any modification neither to the loss function nor to the algorithm itself, as opposed to the conventional techniques used in robust statistics [Huber, 2004]. Finally, we argue that our work has potential implications in constructing confidence intervals in the infinite noise variance setting.

## 2   Preliminaries and Technical Background

**Notational Conventions.** By $\mathbb{N}$, $\mathbb{N}^+$ and $\mathbb{R}$ we denote the set of non-negative integers, positive integers, and real numbers, respectively. For $m \in \mathbb{N}^+$, we define $[m] = \{1, \dots, m\}$. We use italic letters (e.g. $x, \xi$) to denote scalars and scalar-valued functions, $\text{sign}(x)$ to denote the sign of $x$, bold face italic letters (e.g. $\boldsymbol{x}, \boldsymbol{\xi}$) to denote vectors and vector-valued functions, and bold face upper case letters (e.g. $\mathbf{A}$) to denote matrices. We use $|\boldsymbol{x}|$ and $\|\boldsymbol{x}\|_p$ to denote the 2-norm and $p$-norm of a vector $\boldsymbol{x}$; $\|\mathbf{A}\|$ and $\|\mathbf{A}\|_p$ the operator 2-norm and operator $p$-norm of a matrix $\mathbf{A}$. The transpose

of a matrix $\mathbf{A}$ and a vector $\boldsymbol{x}$ (viewed as a matrix with 1 column) are denoted by $\mathbf{A}^\mathsf{T}$ and $\boldsymbol{x}^\mathsf{T}$. If $\{\mathbf{A}_i\}_{i\in\mathbb{N}}$ is a sequence of matrices and $k > \ell$, the empty product $\prod_{i=k}^\ell \mathbf{A}_i$ is understood to be the identity matrix $\mathbf{I}$. For two sequences of real numbers $\{a_t\}_{t\in\mathbb{N}}$, $\{b_t\}_{t\in\mathbb{N}}$, we write $a_t = \mathcal{O}(b_t)$ if $\limsup_{t\to\infty} |a_t|/|b_t| < \infty$, $a_t = o(b_t)$ if $\limsup_{t\to\infty} |a_t|/|b_t| = 0$, $a_t = \Theta(b_t)$ if both $a_t = \mathcal{O}(b_t)$ and $b_t = \mathcal{O}(a_t)$ hold, and $a_t \asymp b_t$ if $\lim_{t\to\infty} |a_t|/|b_t|$ exists and is in $(0, \infty)$. If $a_t = \mathcal{O}(b_t t^\varepsilon)$ for any $\varepsilon > 0$, we say $a_t = \widetilde{\mathcal{O}}(b_t)$. Sufficiently large or sufficiently small positive constants whose values do not matter are written as $C, C_0, C_1, \ldots$, sometimes without prior introduction. If $\boldsymbol{X}_1, \boldsymbol{X}_2, \ldots$ is a sequence of random vectors taking value in $\mathbb{R}^n$ and $\mu$ is a probability measure on $\mathbb{R}^n$, we write $\boldsymbol{X}_t \xrightarrow[t\to\infty]{\mathcal{D}} \mu$ if $\{\boldsymbol{X}_t\}_{t\in\mathbb{N}^+}$ converges in distribution (also called 'weak convergence') to $\mu$.

**Stochastic Approximation.** In the SGD recursion (1.2), we can replace $\boldsymbol{\nabla} f$ with an arbitrary continuous function $\boldsymbol{R} : \mathbb{R}^n \to \mathbb{R}^n$, and consider the same iterations that stochastically approximate the zero $\boldsymbol{x}^*$ of $\boldsymbol{R}$,

$$\boldsymbol{x}_{t+1} = \boldsymbol{x}_t - \gamma_{t+1}\big(\boldsymbol{R}(\boldsymbol{x}_t) + \boldsymbol{\xi}_{t+1}(\boldsymbol{x}_t)\big). \tag{2.1}$$

This is called the *stochastic approximation* process [Robbins and Monro, 1951], which is a predecessor of stochastic gradient descent (SGD) and describes a larger family of iterative algorithms (see e.g. [Kushner and Yin, 2003, Chapters 2 and 3]). Theoretical investigation of the recursion (2.1) has been active ever since its invention, especially under finite noise variance assumption: Robbins and Monro [1951] prove that the recursion (2.1) can lead to the $L^2$ convergence $\lim_{t\to\infty} \mathbb{E}[|\boldsymbol{x}_t - \boldsymbol{x}^*|^2] = 0$; Chung [1954] further calculates an exact convergence rate (see (3.6) in Section 3); Blum [1954] presents an elegant proof that the convergence of $\boldsymbol{x}_t$ to $\boldsymbol{x}^*$ can hold almost surely. The asymptotic distribution of (2.1) can be attributed to [Chung, 1954, Theorem 6], which states that the expression $\gamma_t^{-1/2}(\boldsymbol{x}_t - \boldsymbol{x}^*)$ converges weakly to a normal distribution. In their seminal works, Polyak and Juditsky [1992] and Ruppert [1988] independently introduce the concept of 'averaging the iterates',

$$\overline{\boldsymbol{x}}_t = \frac{\boldsymbol{x}_0 + \ldots + \boldsymbol{x}_{t-1}}{t},$$

showing the striking result that $\sqrt{t}(\overline{\boldsymbol{x}}_t - \boldsymbol{x}^*)$ converges weakly to a fixed normal distribution *regardless of the choice of the step-size* $\{\gamma_t\}_{t\in\mathbb{N}^+}$ as long as it satisfies mild conditions. Recently, optimization algorithms that can handle heavy-tailed noise sequence $\{\boldsymbol{\xi}_t\}_{t\in\mathbb{N}^+}$ have been proposed [Davis et al., 2019, Nazin et al., 2019, Gorbunov et al., 2020]; however, they still rely on a *uniformly* bounded variance assumption, hence do not cover our setting.

Compared with the copious collection of theoretical studies on stochastic approximation algorithms with finite variance as mentioned above, papers that study stochastic approximation under *infinite* noise variance are extremely scarce; we shall summarize only a few papers known to us. Krasulina [1969] is the first to consider such problems, proving almost sure and $L^p$ convergence for the one-dimensional stochastic approximation process without variance. The weak convergence of the iterates (without averaging) $t^{1/\alpha}(\boldsymbol{x}_t - \boldsymbol{x}^*)$ is also considered by Krasulina [1969], but only for the fastest-decaying step-size $\gamma_t = 1/t$. Goodsell and Hanson [1976] discuss how $\boldsymbol{x}_t \to \boldsymbol{x}^*$ in probability can imply $\boldsymbol{x}_t \to \boldsymbol{x}^*$ almost surely, when no finite variance is assumed, and Li [1994] provides a necessary and sufficient condition for almost sure convergence of $\boldsymbol{x}_t \to \boldsymbol{x}^*$, stating that faster-decaying step-size $\gamma_t = o(t^{-1/p})$ is required when moments of lower orders $\mathbb{E}[|\boldsymbol{\xi}_t|^p]$ are not in place. Anantharam and Borkar [2012] show that although step-size that decays slower than $t^{-1/p}$ cannot yield almost sure convergence, $L^p$ convergence can still hold under what they call the 'stability assumption', but their analysis technique provides no convergence rate. Recently, Şimşekli et al. [2019] and Zhang et al. [2020] considered SGD with heavy-tailed noise $\boldsymbol{\xi}_t$ having *uniformly bounded* $p$-th order moments. Besides not being able to handle state-dependent noise due to this uniform moment condition, Şimşekli et al. [2019] imposed further conditions on $\boldsymbol{R} = \boldsymbol{\nabla} f$ such as global Hölder continuity for a non-convex $f$, whereas Zhang et al. [2020] modified SGD with 'gradient clipping', in order to be able to compensate the effects of the heavy-tailed noise.

Finally, we shall mention that a class of stochastic recursions similar to (2.1) have been considered in the dynamical systems theory [Mirek, 2011, Buraczewski et al., 2012, 2016], for which generalized central limit theorems with $\alpha$-stable limits have been established. However, such techniques typically require $\boldsymbol{R}$ to be (asymptotically) linear and the step-sizes to be constant as they heavily rely on the theory of time-homogeneous Markov processes. Hence, their approach does not readily generalize to the setting of our interest, i.e., non-linear $\boldsymbol{R}$ and diminishing step-sizes, where the latter is crucial for ensuring convergence towards the global optimum.

**Stable Distributions.** In probability theory, a random variable $X$ is *stable* if its distribution is non-degenerate and satisfies the following property: Let $X_1$ and $X_2$ be independent copies of $X$. Then, for any constants $a, b > 0$, the random variable $aX_1 + bX_2$ has the same distribution as $cX + d$ for some constants $c > 0$ and $d$ (see e.g. [Samorodnitsky and Taqqu, 1994]). The stable distribution is also referred to as the $\alpha$-stable distribution, first proposed by Lévy [1937], where $\alpha \in (0, 2]$ denoting the stability parameter. The case $\alpha = 2$ corresponds to the normal distribution, and the variance under this distribution is undefined for any $\alpha < 2$. The multivariate $\alpha$-stable distribution dates back to Feldheim [1937], which is a multivariate generalization of the univariate $\alpha$-stable distribution, which is also uniquely characterized by its characteristic function. In particular, an $\mathbb{R}^n$-valued random vector $X$ has a multivariate $\alpha$-stable distribution, denoted as $\boldsymbol{X} \sim \mathcal{S}(\alpha, \Lambda, \delta)$ if the joint characteristic function of $\boldsymbol{X}$ is given by

$$\mathbb{E}\big[\exp\big(i\boldsymbol{u}^\mathsf{T}\boldsymbol{X}\big)\big] = \exp\Big\{ -\int_{\boldsymbol{s}\in S_2}(|\boldsymbol{u}^\mathsf{T}\boldsymbol{s}|^\alpha + i\nu(\boldsymbol{u}^\mathsf{T}\boldsymbol{s}, \alpha))\Lambda(\mathrm{d}\boldsymbol{s}) + i\boldsymbol{u}^\mathsf{T}\delta\Big\},$$

for any $\boldsymbol{u} \in \mathbb{R}^n$, and $0 < \alpha \leqslant 2$. Here, $\alpha$ is the tail-index, $\Lambda$ is a finite measure on $S_2$ known as the spectral measure, $\boldsymbol{\delta} \in \mathbb{R}^n$ is a shift vector, and $\nu(y, \alpha) := -\operatorname{sign}(y)\tan(\pi\alpha/2)|y|^\alpha$ for $\alpha \neq 1$ and $\nu(y, \alpha) := (2/\pi)y\log|y|$ for $\alpha = 1$ for any $y \in \mathbb{R}$, and $S_2$ denotes the unit sphere in $\mathbb{R}^n$; i.e. $S_2 = \{\boldsymbol{s} \in \mathbb{R}^n : |\boldsymbol{s}| = 1\}$. Stable distribution also emerges as the limit distribution in the Generalized Central Limit Theorem (GCLT) [Gnedenko and Kolmogorov, 1954], which states that for a sequence of i.i.d. random variables whose distributions have a power-law tail with index $0 < \alpha < 2$, the normalized sum converges to an $\alpha$-stable distribution as the number of summands go to $\infty$.

**Domains of Normal Attraction of Stable Distributions.** Let $\boldsymbol{X}_1, \boldsymbol{X}_2, \ldots, \boldsymbol{X}_t$ be an i.i.d. sequence of random vectors in $\mathbb{R}^n$ with a common distribution function $F(\boldsymbol{x})$. If there exists some constant $a > 0$ and a sequence $b_t \in \mathbb{R}^n$ such that

$$\frac{\boldsymbol{X}_1 + \cdots + \boldsymbol{X}_t}{at^{1/\alpha}} - b_t \xrightarrow[t\to\infty]{\mathcal{D}} \mu, \tag{2.2}$$

then $F(\boldsymbol{x})$ is said to belong to the *domain of normal attraction* of the law $\mu$, and $\alpha$ is the characteristic exponent of $\mu$ [Gnedenko and Kolmogorov, 1954, page 181]. If $\mu$ is an $\alpha$-stable distribution, then we say $F(\boldsymbol{x})$ belongs to the domain of normal attraction of an $\alpha$-stable distribution. For example, the Pareto distribution belongs to the domain of normal attraction of an $\alpha$-stable law. In Section C in the supplementary document, we provide more details as well as a sufficient and necessary conditions for being in the domain of normal attraction of an $\alpha$-stable law.

## 3 Convergence of SGD under Heavy-tailed Gradient Noise

In this section, we identify sufficient conditions for the convergence of SGD under heavy-tailed gradient noise, and derive explicit rate estimates. In the standard setting when the noise variance is finite, it is sufficient to assume Hessian is uniformly positive definite in order to achieve contraction in the subsequent SGD iterations (see for example Polyak and Juditsky [1992], Tripuraneni et al. [2018], Su and Zhu [2018], Duchi and Ruan [2021], Toulis and Airoldi [2017], Fang et al. [2018], Anastasiou et al. [2019]). When the noise variance is infinite with a finite $p$-th moment for $p \in [1, 2)$, a stronger notion of positive definiteness is required in our analysis to achieve such a contraction, which leads to an interesting interpolation between the positive semi-definite cone (as $p \to 2$), and the cone of diagonally dominant matrices with non-negative diagonal entries ($p = 1$).

### 3.1 $p$-Positive Definiteness

First, we introduce the signed power of vectors which will be used to define a family of matrices.

For a vector $\boldsymbol{v} = (v^1, \ldots, v^n)^\mathsf{T} \in \mathbb{R}^n$ and $q \geqslant 0$, the signed power of $\boldsymbol{v}$ is defined as

$$\boldsymbol{v}^{\langle q\rangle} := \Big(\operatorname{sign}\big(v^1\big)\big|v^1\big|^q, \ldots, \operatorname{sign}(v^n)|v^n|^q\Big)^\mathsf{T}. \tag{3.1}$$

Denoting the $n$-dimensional $\ell_p$ unit sphere with $S_p = \{\boldsymbol{v} \in \mathbb{R}^n : \|\boldsymbol{v}\|_p = 1\}$, and the set of $n \times n$ symmetric matrices with $\mathbb{S}$, we now define the following subset of $\mathbb{S}$.

**Definition 1** ($p$-positive definiteness)**.** *Let $p \geqslant 1$ and $\mathbf{Q}$ be a symmetric matrix. We say that $\mathbf{Q}$ is $p$-positive definite if for all $\boldsymbol{v} \in S_p$, we have $\boldsymbol{v}^\mathsf{T}\mathbf{Q}\boldsymbol{v}^{\langle p-1\rangle} > 0$. Similarly, we say that $\mathbf{Q}$ is $p$-positive semi-definite if for all $\boldsymbol{v} \in S_p$, we have $\boldsymbol{v}^\mathsf{T}\mathbf{Q}\boldsymbol{v}^{\langle p-1\rangle} \geqslant 0$.*

It is not hard to see that the set of $p$-positive semi-definite matrices ($p$-PSD) defines a closed pointed cone, which we denote by $\mathbb{S}_+^p$, with interior as the set of $p$-positive definite matrices ($p$-PD), denoted by $\mathbb{S}_{++}^p$. We are mainly interested in the case $1 \leqslant p < 2$. Note that $\mathbb{S}_+^2$ coincides with the standard PSD cone, and we show in Section A.2 that $\mathbb{S}_+^1$ is exactly the cone of diagonally dominant matrices with non-negative diagonal entries, denoted by $\mathbb{D}_+$. For any $p \in [1, 2]$, these cones satisfy the following

$$\mathbb{D}_+ = \mathbb{S}_+^1 \subseteq \mathbb{S}_+^p \subseteq \mathbb{S}_+^2.$$

Figure 1 is a hypothetical illustration of the inclusion relationship between these cones.

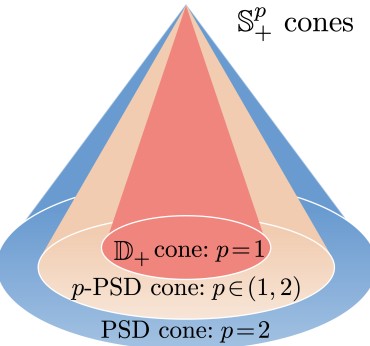

Figure 1: Geometry of $p$-PSD matrices. $\mathbb{D}_+$ cone refers to the cone of diagonally dominant matrices with non-negative diagonal entries. Their inclusion relationship is given in Propositions 13 and 14.

Similar to the uniform PD condition on the Hessian, which is commonly used in classical analysis (i.e. strong convexity), we also define a uniform version of Definition 1. We recall that every operator norm $\|\cdot\|_p$ induces the same topology on the set of $n$-dimensional matrices, which is just the usual topology on $\mathbb{R}^{n \times n}$. Further, the set of symmetric matrices $\mathbb{S}$, as the set of zeros of the continuous function $\mathbf{X} \mapsto \mathbf{X} - \mathbf{X}^\mathsf{T}$, is a closed set. Hence for a set $\mathcal{M} \subseteq \mathbb{S}$, denoting its topological closure with $\overline{\mathcal{M}}$, we also have $\overline{\mathcal{M}} \subseteq \mathbb{S}$. We are interested in the case where $\mathcal{M}$ is a bounded set.

**Definition 2** (uniform $p$-PD). *Let $p \geqslant 1$ and $\mathcal{M} \subset \mathbb{S}$ be a non-empty set of symmetric matrices. We say that $\mathcal{M}$ is uniformly $p$-PD if for all $\mathbf{Q} \in \overline{\mathcal{M}}$, we have $\mathbf{Q} \in \mathbb{S}_{++}^p$.*

Notice that $\mathcal{M}$ is uniformly 2-PD if and only if the eigenvalues of the symmetric matrices in the set $\mathcal{M}$ are all lower bounded by a positive real number. Notice also that a finite subset of symmetric matrices is uniformly $p$-PD if and only if each element of the set is $p$-PD.

$p$-PSD cone emerges naturally when analyzing SGD algorithm in the heavy-tailed setting, interpolating between the standard PSD cone to the cone of diagonally dominant matrices with non-negative diagonal entries. To the best of our knowledge, we are the first to study such families of matrices and their application in stochastic optimization. For further details about these cones, we refer interested reader to Section A.2 in the supplementary document.

We make the following uniform smoothness and curvature assumptions on the objective function.

**Assumption 1.** *The set of matrices $\{\boldsymbol{\nabla}^2 f(\boldsymbol{x}) : \boldsymbol{x} \in \mathbb{R}^n\}$ is bounded and uniformly $p$-PD.*

### 3.2 Rate of Convergence in $L^p$

We fix a probability space $(\Omega, \mathcal{F}, \mathbb{P})$ with filtration $\{\mathcal{F}_t\}_{t \in \mathbb{N}}$, and we let $\boldsymbol{x}_0$ be $\mathcal{F}_0$-measurable. We make the following assumption on the gradient noise sequence.

**Assumption 2.** *The gradient noise sequence $\{\boldsymbol{\xi}_t\}_{t \in \mathbb{N}^+}$ admits the following decomposition*

$$\boldsymbol{\xi}_{t+1}(\boldsymbol{x}_t) = \boldsymbol{m}_{t+1}(\boldsymbol{x}_t) + \boldsymbol{\zeta}_{t+1}, \tag{3.2}$$

*where $\{\boldsymbol{\zeta}_t\}_{t \in \mathbb{N}^+}$ is an i.i.d. sequence with $\mathbb{E}[\boldsymbol{\zeta}_t] = 0$, and $\mathbb{E}[|\boldsymbol{\zeta}_t|^p] < \infty$ for some $p$, and $\{\boldsymbol{m}_t\}_{t \in \mathbb{N}^+}$ is a martingale difference sequence, and both sequences are adapted to the filtration $\{\mathcal{F}_t\}_{t \in \mathbb{N}}$.*

*Further, the state dependent component of the noise satisfies, for some $K > 0$,*

$$\mathbb{E}\left[|\boldsymbol{m}_{t+1}(\boldsymbol{x}_t)|^2 \mid \mathcal{F}_t\right] \leqslant K(1 + |\boldsymbol{x}_t|^2). \tag{3.3}$$

We call $\boldsymbol{m}_t$ the *state-dependent component* of the gradient noise, which naturally has a state-dependent conditional second moment. The variance of this component can be arbitrarily large depending on the state $\boldsymbol{x}_t$. The *heavy-tailed* noise behavior is due to $\boldsymbol{\zeta}_t$, which may have an infinite variance for $p < 2$ (i.e., the second moment is undefined). Compared to recent works on SGD with heavy-tailed noise, our noise model in Assumption 2 is significantly more general. For instance, the noise model in the recent work Zhang et al. [2020] assumes $\mathbb{E}[|\boldsymbol{\xi}_{t+1}(\boldsymbol{x})|^p] \leq \sigma^p$ for all $\boldsymbol{x}$, where $\sigma$ does not depend on $\boldsymbol{x}$. Therefore, this noise model cannot handle state-dependent noise, and does not even hold in the

linear regression example given in Section 5 (as the moments of the noise must scale with the norm of $\boldsymbol{x}$). On the contrary, in many instances of stochastic approximation methods subject to heavy-tailed noise with long-range dependencies, one can verify that the noise admits the decomposition (3.2). We shall show in Section 5 that the noise model (3.2) arises in practical applications such as linear regression and generalized linear models subject to heavy-tailed data (see also Anantharam and Borkar [2012] for a detailed discussion on this noise model).

In our first result, assuming that the objective function $f$ has a uniformly $p$-PD Hessian and the noise sequence $\{\boldsymbol{\xi}_t\}_{t\in\mathbb{N}^+}$ has infinite variance but satisfies Assumption 2, we establish an asymptotic convergence rate in $L^p$ for the SGD algorithm to the unique minimizer $\boldsymbol{x}^*$.

**Theorem 3.** *Suppose Assumptions 1 and 2 hold for some $1 < p \leqslant 2$. For step-size satisfying $\gamma_t \asymp t^{-\rho}$ with $\rho \in (0,1)$, the error of the SGD iterates $\{\boldsymbol{x}_t\}_{t\in\mathbb{N}}$ from the minimizer $\boldsymbol{x}^*$ satisfies*

$$\mathbb{E}[|\boldsymbol{x}_t - \boldsymbol{x}^*|^p] = \mathcal{O}\Big(t^{-\rho(p-1)}\Big). \tag{3.4}$$

*Consequently, we have $\sup_{t\in\mathbb{N}^+} \mathbb{E}[|\boldsymbol{\xi}_t|^p] < \infty$.*

The proof of Theorem 3 is provided in Section B in the supplementary document. We observe that the convergence rate of SGD depends on the highest order finite moment $p$ of the noise sequence, and faster rates are achieved for larger values of $p$. The fastest convergence rate implied by our result is near $\mathcal{O}\big(t^{-p+1}\big)$, which is achieved for $\rho \approx 1$. However, SGD converges even for very slowly decaying step-size sequences as $\rho$ gets closer to $0$.

If the noise sequence has further integrability properties with a finite $p$-th moment for all $p \in [q, \alpha)$ for some $1 < q < \alpha$ and if uniform $p$-PD condition (i.e. Assumption 1) holds, then faster rates are achievable. In particular, the following result is an interesting consequence of Theorem 3, and its proof is provided in Section B in the supplementary document.

**Corollary 4.** *For constants $q, \alpha$ satisfying $1 < q < \alpha \leqslant 2$, suppose that Assumptions 1 and 2 hold for every $p \in [q, \alpha)$. For step-size satisfying $\gamma_t \asymp t^{-\rho}$ with $\rho \in (0,1)$, the error of the SGD iterates $\{\boldsymbol{x}_t\}_{t\in\mathbb{N}}$ from the minimizer $\boldsymbol{x}^*$ satisfies*

$$\mathbb{E}[|\boldsymbol{x}_t - \boldsymbol{x}^*|^q] = \tilde{\mathcal{O}}\Big(t^{-\rho q \frac{\alpha-1}{\alpha}}\Big). \tag{3.5}$$

**Remark.** The additional integrability assumption yields faster rates for any feasible step-size sequence since $p(\alpha - 1)/\alpha \geqslant p - 1$ for $p \in (1, 2]$.

Let us briefly compare our results stated above to those in the setting where the noise sequence has a finite variance. A classical convergence result that goes back to Chung [1954, Theorem 5][2] states that

$$\mathbb{E}[|\boldsymbol{x}_t - \boldsymbol{x}^*|^r] = \Theta\Big(t^{-\rho r/2}\Big), \tag{3.6}$$

where $r \geqslant 2$ is an integer such that the $r$-th moment exists for the stochastic approximation process, and this is achieved for strongly convex objective functions in one dimension (whose second derivative $\{f''(\boldsymbol{x}) : \boldsymbol{x} \in \mathbb{R}\}$ satisfies the uniformly 2-PD property) with a step-size choice $\gamma_t \asymp t^{-\rho}$ for some $\rho \in (1/2, 1)$. We point out that our rate (3.5) recovers the rate implied by (3.6) when $r = 2$, and extends it further to the case $1 \leqslant r < 2$.

In the presence of heavy-tailed noise, the folklore is to modify SGD (e.g., clipped gradients) in order to tame the heavy-tails, which considerably simplifies the problem and makes it amenable to classical analysis tools. For instance, to motivate modifying SGD in this regime, in [Zhang et al., 2020, Remark 1] authors prove that $\mathbb{E}\big[|\nabla f(\boldsymbol{x}_t)|^2\big] = \infty$ for vanilla SGD and argue that SGD diverges in this setting. On the contrary, our results show that, without any modifications, SGD can still converge to the optimum in $L^p$, even when it does not converge in $L^2$ since the second moment is not defined.

## 4  Stable Limits for the Polyak-Ruppert Averaging

In this section, we establish the limit distribution of the Polyak-Ruppert averaging under infinite noise variance, extending the asymptotic normality result given by Polyak and Juditsky [1992] to $\alpha$-stable distributions. Let us fix an $\alpha \in (1, 2]$ and assume the following throughout this subsection.

---

[2]This result, like many other similar studies in the 1950s, concerns only the one-dimensional case. But they generalize easily to higher dimensions.

**Assumption 3.** *The gradient noise sequence $\{\boldsymbol{\xi}_t\}_{t\in\mathbb{N}^+}$ admits the following decomposition*

$$\boldsymbol{\xi}_{t+1}(\boldsymbol{x}_t) = \boldsymbol{m}_{t+1}(\boldsymbol{x}_t) + \boldsymbol{\zeta}_{t+1}, \tag{4.1}$$

*where $\{\boldsymbol{\zeta}_t\}_{t\in\mathbb{N}^+}$ is an i.i.d. sequence with $\mathbb{E}[\boldsymbol{\zeta}_t] = 0$, which is in the domain of normal attraction of an $n$-dimensional symmetric $\alpha$-stable distribution $\mu$, i.e.,*

$$\frac{\boldsymbol{\zeta}_1 + \ldots + \boldsymbol{\zeta}_t}{t^{1/\alpha}} \xrightarrow[t\to\infty]{\mathcal{D}} \mu. \tag{4.2}$$

*The state dependent component $\{\boldsymbol{m}_t\}_{t\in\mathbb{N}^+}$ is a martingale difference sequence with a second-moment satisfying* (3.3), *and both sequences are adapted to the filtration $\{\mathcal{F}_t\}_{t\in\mathbb{N}}$.*

The above assumption is arguably more stringent than Assumption 2, and it implies that $\mathbb{E}[|\boldsymbol{\zeta}_t|^p] < \infty$ for all $p \in [1, \alpha)$; thus, Assumption 2 is satisfied for any $p \in [1, \alpha)$. The condition (4.2) is a special case of (2.2) which holds if, for example, $\boldsymbol{\zeta}_t$'s have power-law tails (i.e. Paretian tail) with index $\alpha$.

Denoting the Polyak-Ruppert averaging by $\overline{\boldsymbol{x}}_t := \frac{1}{t}(\boldsymbol{x}_0 + \ldots + \boldsymbol{x}_{t-1})$, we are interested in the asymptotic behavior of

$$t^{1-1/\alpha}(\overline{\boldsymbol{x}}_t - \boldsymbol{x}^*) = \frac{(\boldsymbol{x}_0 + \ldots + \boldsymbol{x}_{t-1}) - t\boldsymbol{x}^*}{t^{1/\alpha}},$$

for $\alpha \in (1, 2]$. In the special case when $\alpha = 2$, it is known that this limit converges to a multivariate normal distribution (which is a 2-stable distribution), a result proven in the seminal work by Polyak and Juditsky [1992]. Similarly, we begin with a result that considers a quadratic objective where the function $\boldsymbol{\nabla} f(\boldsymbol{x})$ is linear in $\boldsymbol{x}$, and then building on this result, we establish the limit distribution of Polyak-Ruppert averaging also in the more general non-linear case.

**Theorem 5** (linear case). *Suppose the function $\boldsymbol{\nabla} f(\boldsymbol{x})$ is affine, i.e. $\boldsymbol{\nabla} f(\boldsymbol{x}) = \mathbf{A}\boldsymbol{x} - \boldsymbol{b}$ for a real matrix $\mathbf{A} \in \mathbb{R}^{n \times n}$ and a real vector $\boldsymbol{b} \in \mathbb{R}^n$ and there exist scalars $p, \rho$ satisfying*

$$\max\left(\frac{\alpha + \alpha\rho}{1 + \alpha\rho}, \alpha\rho\right) \leqslant p \leqslant \alpha,$$

*such that $\mathbf{A}$ is $p$-PD and $\rho \in (0, 1)$. If the noise sequence satisfies Assumption 3 for the same parameter $\alpha$, then for the step-size satisfying $\gamma_t \asymp t^{-\rho}$, the normalized average $t^{1-1/\alpha}(\overline{\boldsymbol{x}}_t - \boldsymbol{x}^*)$ converges weakly to an $n$-dimensional $\alpha$-stable distribution.*

The above theorem states that Polyak-Ruppert averaging for any step-size sequence with index $\rho \in (0, 1]$ converges weakly to an $\alpha$-stable limit. Thus, in the linear case, the size of this feasible interval is the same in both heavy- and light-tailed noise settings (see e.g. Polyak and Juditsky [1992] and Ruppert [1988]). Notably, $\alpha$-stable limit of the averaged iterates does not depend on the index $\rho$, i.e., limit distribution does not depend on how fast the step-size decays as long as $\rho \in (0, 1]$.

The next result generalizes Theorem 5 to the setting where $\boldsymbol{\nabla} f(\boldsymbol{x})$ is non-linear.

**Theorem 6** (non-linear case). *Let $1 < 1/\rho < q < \alpha$ and suppose Assumption 1 holds for every $p \in [q, \alpha)$. Assume further that the gradient $\boldsymbol{\nabla} f(\boldsymbol{x})$ can be approximated using the Hessian matrix $\boldsymbol{\nabla}^2 f(\boldsymbol{x}^*)$ around the minimizer $\boldsymbol{x}^*$ as*

$$\left|\boldsymbol{\nabla} f(\boldsymbol{x}) - \boldsymbol{\nabla}^2 f(\boldsymbol{x}^*)(\boldsymbol{x} - \boldsymbol{x}^*)\right| \leqslant K|\boldsymbol{x} - \boldsymbol{x}^*|^q. \tag{4.3}$$

*If the noise sequence satisfies Assumption 3, for the step-size satisfying $\gamma_t \asymp t^{-\rho}$, the normalized average $t^{1-1/\alpha}(\overline{\boldsymbol{x}}_t - \boldsymbol{x}^*)$ converges weakly to an $n$-dimensional $\alpha$-stable distribution.*

The condition (4.3) is standard (see e.g. Polyak and Juditsky [1992, Assumption 3.2]), which simply imposes a local linearity condition on the gradient of the objective function $f$, with an order-$q$ polynomial error term. This assumption holds, for example, whenever the Hessian is Lipschitz continuous. We notice that the size of the feasible interval is $\rho \in (1/\alpha, 1)$, which is smaller this time compared to the light tailed case; Polyak and Juditsky [1992, Theorem 2] allows $\rho \in (1/2, 1)$.

The above theorem establishes that, when the noise has diverging variance, the Polyak-Ruppert averaging admits an $\alpha$-stable limit rather than a standard CLT. This result has potential implications in statistical inference in the presence of heavy-tailed data. Inference procedures that take into account the computational part of the training procedure (instead of drawing conclusions for the minimizer

of the empirical risk) rely typically on variations of Polyak-Ruppert averaging and the CLT they admit [Fang et al., 2018, Su and Zhu, 2018, Chen et al., 2020]. The above theorem simply states this CLT does not hold under heavy-tailed gradient noise. Therefore, many of these procedures require further adaptation, if the gradient has undefined variance. Finally, it is well-known that Polyak-Ruppert averaging achieves the Cramér-Rao lower bound [Polyak and Juditsky, 1992, Gadat and Panloup, 2017], which is a lower bound on the variance of an unbiased estimator. However, it is not clear what this type of optimality means when the variance is not defined. These are important directions that require thorough investigations, and they will be studied elsewhere.

## 5 Examples in the Presence of Heavy-tailed Noise

In this section, we demonstrate how the stochastic approximation framework discussed in our paper covers several interesting examples. More specifically, we verify the assumptions required for Theorems 3, 5, and 6, for linear regression and generalized linear models (GLMs), where the heavy-tailed noise behavior may naturally arise due to heavy-tailed data.

### 5.1 Ordinary Least Squares

Let us first consider the following linear model,

$$y = \boldsymbol{z}^\mathsf{T} \boldsymbol{\beta}_0 + \epsilon,$$

where $\boldsymbol{\beta}_0 \in \mathbb{R}^n$ is the true coefficients, $y \in \mathbb{R}$ is the response, the random vector $\boldsymbol{z} \in \mathbb{R}^n$ denotes the covariates with a positive-definite covariance $0 \prec \mathbb{E}[\boldsymbol{z}\boldsymbol{z}^\mathsf{T}]$ and a finite fourth moment $\mathbb{E}[|\boldsymbol{z}|^4] < \infty$, and $\epsilon$ is the noise with zero conditional mean $\mathbb{E}[\epsilon|\boldsymbol{z}] = 0$. In the classical setting, the noise $\epsilon$ is assumed to be Gaussian whose variance is well-defined. In this case, the population version of the maximum likelihood estimation (MLE) problem corresponds to minimizing

$$f(\boldsymbol{x}) = \frac{1}{2}\mathbb{E}\Big[\big(y - \boldsymbol{z}^\mathsf{T}\boldsymbol{x}\big)^2\Big], \tag{5.1}$$

(where the expectation is over the $(y, \boldsymbol{z})$ pair), or equivalently solving the following normal equations

$$\boldsymbol{\nabla} f(\boldsymbol{x}) := \mathbb{E}\big[\boldsymbol{z}\boldsymbol{z}^\mathsf{T}\big]\boldsymbol{x} - \mathbb{E}[\boldsymbol{z}y] = 0. \tag{5.2}$$

We easily observe that the true coefficients $\boldsymbol{\beta}_0$ is the unique zero of the above equation, i.e., $\boldsymbol{x}^* = \boldsymbol{\beta}_0$.

Now, suppose we are given access to a stream of i.i.d. drawn instances of the pair $(y, \boldsymbol{z})$, denoted by $\{y_t, \boldsymbol{z}_t\}_{t \in \mathbb{N}^+}$. In large-scale settings, one generally runs the following stochastic approximation process, which is simply online SGD on the population MLE objective $f(\boldsymbol{x})$:

$$\boldsymbol{x}_t = \boldsymbol{x}_{t-1} - \gamma_t\big(\boldsymbol{z}_t\boldsymbol{z}_t^\mathsf{T}\boldsymbol{x}_{t-1} - \boldsymbol{z}_t y_t\big). \tag{5.3}$$

Manifestly, (5.3) is a special case of (2.1), where the gradient noise admitting the decomposition $\boldsymbol{\xi}_t = \boldsymbol{\zeta}_t + \boldsymbol{m}_t$, for an i.i.d. component $\boldsymbol{\zeta}_t$ and a state-dependent component $\boldsymbol{m}_t$ (cf. (4.1)),

$$\begin{cases} \boldsymbol{\zeta}_t := \mathbb{E}[\boldsymbol{z}y] - \boldsymbol{z}_t y_t, \\ \boldsymbol{m}_t := \big(\boldsymbol{z}_t\boldsymbol{z}_t^\mathsf{T} - \mathbb{E}\big[\boldsymbol{z}\boldsymbol{z}^\mathsf{T}\big]\big)\boldsymbol{x}_{t-1}. \end{cases} \tag{5.4}$$

In the presence of heavy-tailed noise, for example when the noise $\epsilon$ has infinite variance, the population MLE objective $f(\boldsymbol{x})$ may not be even finite, and one should resort to methods from M-estimation and choose a suitable loss function within robust statistics framework [Huber, 2004, Van der Vaart, 2000]. However, the iterations (5.3) may still be employed to estimate the true coefficients $\boldsymbol{\beta}_0$ (potentially due to model misspecification), as we demonstrate below. Note that in this case, iterations (5.3) should be seen as solving the root-finding problem (5.2) via stochastic approximation (2.1), rather than a minimization problem stated in (5.1).

First, we verify Assumption 2. We observe from the decomposition given in (5.4) that the i.i.d. component $\{\boldsymbol{\zeta}_t\}_{t \in \mathbb{N}}$ exhibits the heavy-tailed behavior since it contains $y_t = \boldsymbol{z}_t^\mathsf{T}\boldsymbol{\beta}_0 + \epsilon_t$. Assume that this component has the highest order finite moment $p$ satisfying $1 \leqslant p < 2$, i.e., $\mathbb{E}[|\boldsymbol{\zeta}_t|^p] < \infty$. Further, the state dependent component $\boldsymbol{m}_t$ defines a martingale difference sequence, and the condition (3.3) is met since the covariates $\boldsymbol{z}$ have finite fourth moment, i.e.,

$$\mathbb{E}\big[|\boldsymbol{m}_t|^2 \mid \boldsymbol{x}_{t-1}\big] \leqslant C|\boldsymbol{x}_{t-1}|^2.$$

Hence, Assumption 2 is satisfied. Next, assuming that the second moment of the covariates $\nabla^2 f(\boldsymbol{x}) = \mathbb{E}[\boldsymbol{z}\boldsymbol{z}^\mathsf{T}]$ is $p$-PD, one can guarantee that Assumption 1 is satisfied. Therefore, the convergence results of our theorems hold. Finally, we note that $p$-PD assumption is always satisfied if $\mathbb{E}[\boldsymbol{z}\boldsymbol{z}^\mathsf{T}]$ is diagonally dominant, but the condition for $p > 1$ is weaker.

## 5.2 Generalized Linear Models

Generalized linear models (GLMs) play a crucial role in numerous problems in statistics, and provide a miscellaneous framework for many regression and classification tasks, with many applications [McCullagh and Nelder, 1989, Nelder and Wedderburn, 1972]. In this section, we consider minimizing the objective function arising from GLMs, for which there are many methods available (see e.g. Erdogdu [2015, 2016] and the references therein). However, we restrict ourselves to the misspecified and online setting. That is, the minimization problem corresponds to a GLM, but the model is misspecified so that the response can be heavy-tailed.

For a response $y \in \mathbb{R}$ and random covariates $\boldsymbol{z} \in \mathbb{R}^n$, the population version of an $\ell_2$-regularized MLE problem in the canonical GLM framework reads

$$\underset{\boldsymbol{x}}{\text{minimize}} \ f(\boldsymbol{x}) := \mathbb{E}\Big[\psi(\boldsymbol{x}^\mathsf{T}\boldsymbol{z}) - y\boldsymbol{x}^\mathsf{T}\boldsymbol{z}\Big] + \frac{\lambda}{2}|\boldsymbol{x}|^2 \qquad \text{for} \qquad \lambda > 0. \tag{5.5}$$

Here, $\psi : \mathbb{R} \to \mathbb{R}$ is referred to as the cumulant generating function (CGF), and it is convex. Notable examples include $\psi(x) = x^2/2$ yielding linear regression, $\psi(x) = \log(1 + e^x)$ yielding logistic regression, and $\psi(x) = e^x$ yielding Poisson regression. Gradient of the objective (5.5) is given by

$$\nabla f(\boldsymbol{x}) = \mathbb{E}\Big[\boldsymbol{z}\psi'(\boldsymbol{z}^\mathsf{T}\boldsymbol{x})\Big] - \mathbb{E}[\boldsymbol{z}y] + \lambda\boldsymbol{x}. \tag{5.6}$$

We define the unique solution of the population GLM problem as the unique zero of (5.6), which we denote by $\boldsymbol{x}^*$. To reiterate, we do not assume a model on data, allowing for model misspecification; we simply consider the resulting optimization problem similar to Erdogdu et al. [2016, 2019]. As in the previous section, we assume that the covariates have finite fourth moment and the response $y_t$ is contaminated with heavy-tailed noise with infinite variance. In this setting, the objective function is always well-defined, even if the response has infinite variance.

We are given access to a stream of i.i.d. drawn instances of the pair $(y, \boldsymbol{z})$, denoted by $\{y_t, \boldsymbol{z}_t\}_{t \in \mathbb{N}^+}$, and we solve the above non-linear problem using the following stochastic process,

$$\boldsymbol{x}_t = \boldsymbol{x}_{t-1} - \gamma_t\big(\boldsymbol{z}_t\psi'(\boldsymbol{z}_t^\mathsf{T}\boldsymbol{x}_{t-1}) - \boldsymbol{z}_t y_t + \lambda\boldsymbol{x}_{t-1}\big),$$

with gradient noise admitting the decomposition $\boldsymbol{\xi}_t = \boldsymbol{\zeta}_t + \boldsymbol{m}_t$ where

$$\begin{cases} \boldsymbol{\zeta}_t := \mathbb{E}[\boldsymbol{z}y] - \boldsymbol{z}_t y_t, \\ \boldsymbol{m}_t := \boldsymbol{z}_t\psi'(\boldsymbol{z}_t^\mathsf{T}\boldsymbol{x}_{t-1}) - \mathbb{E}\big[\boldsymbol{z}_t\psi'(\boldsymbol{z}_t^\mathsf{T}\boldsymbol{x}_{t-1})\big]. \end{cases}$$

In what follows, we verify our assumptions for a CGF satisfying $|\psi'(x)| \leqslant C(1 + |x|)$, $\psi''(x) \geqslant 0$, and $|\psi'''(x)| \leq L$ for all $x \in \mathbb{R}$. These assumptions can be easily verified for any second-order smooth CGF that grows at most quadratically (e.g. if the misspecified model is binomial with $k$ number of trials, we have $\psi(x) = k\log(1 + e^x)$). $\boldsymbol{\zeta}_t$'s are i.i.d. and contain the entire heavy-tailed part of the gradient noise. Assume that this component has the highest defined moment order $1 \leqslant p < 2$, i.e., $\mathbb{E}[|\boldsymbol{\zeta}_t|^p] < \infty$. Further observe that the state dependent component defines a martingale difference sequence and satisfies the condition (3.3) since the covariates $\boldsymbol{z}$ have finite fourth moment, and $|\psi'|$ grows at most linearly. Therefore, Assumption 2 is satisfied.

We note that the Hessian of the objective $f$ is given as

$$\nabla^2 f(\boldsymbol{x}) = \mathbb{E}\big[\boldsymbol{z}\boldsymbol{z}^\mathsf{T}\psi''(\boldsymbol{z}^\mathsf{T}\boldsymbol{x})\big] + \lambda\mathbf{I}.$$

Since $\psi''(x) \geqslant 0$, $\nabla^2 f(\boldsymbol{x})$ is clearly PD for all $\lambda > 0$. For sufficiently large $\lambda$, this matrix can also be made diagonally dominant, which implies the $p$-PD condition for any $p \geqslant 1$, further implying Assumption 1. We further note that if $\nabla^2 f(\boldsymbol{x})$ is Lipshitz (e.g. for the binomail CGF), then (4.3) holds for $q = 2$ globally; thus it holds for any $q < 2$ locally. Therefore, for an appropriate step-size sequence, our convergence results on the SGD can be applied to this framework.

# 6   Conclusion

In this paper, we considered SGD subject to state-dependent and heavy-tailed noise with potentially infinite variance, when the objective belongs to a class of strongly convex functions (termed as $p$-PD condition). We provided asymptotic $L^p$ convergence rates for vanilla SGD, demonstrating that SGD without any modifications can be still used in the presence of heavy-tailed noise. Furthermore, we provided a generalized central limit theorem for the Polyak-Ruppert averaging, i.e., we proved that the averaged iterates converge to a multivariate $\alpha$-stable distribution.

We emphasize that $p$-PD condition is a sufficient condition, and further investigation is needed to see if this condition can be replaced with the standard strong convexity assumption. We also highlight that non-asymptotic $L^p$ rates in the current setting should be achievable, which will be studied elsewhere. Finally, while we leave it for a future study, we emphasized the importance of adapting existing statistical inference techniques that rely on the averaged SGD iterates when the gradient noise is heavy-tailed, which arises naturally in modern statistical learning applications.

## Acknowledgements

MAE is partially funded by CIFAR AI Chairs program, and CIFAR AI Catalyst grant, NSERC Grant [2019-06167]. MG's research is supported in part by the grants Office of Naval Research Award Number N00014-21-1-2244, National Science Foundation (NSF) CCF-1814888, NSF DMS-2053485, NSF DMS-1723085. LZ is grateful to the partial support from NSF DMS-2053454 and a Simons Foundation Collaboration Grant. UŞ's research is supported by the French government under management of Agence Nationale de la Recherche as part of the "Investissements d'avenir" program, reference ANR-19-P3IA-0001 (PRAIRIE 3IA Institute).

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
