# Convergence Rates of Stochastic Gradient Descent under Infinite Noise Variance

## SUPPLEMENTARY DOCUMENT

## A   Lemmas and Discussions

### A.1   Key Lemmas

In this subsection, we present some key lemmas used in the proof of our main theorems, which are helpful when considering stochastic problems with *infinite* variance.

The concept of *uncorrelatedness* has long been used by probabilists as a trick when computing and estimating variance. For example, consider a sequence of uncorrelated random vectors $\{\boldsymbol{X}_t\}_{t\in\mathbb{N}^+}$ (e.g. square-integrable martingale difference). Then

$$\mathbb{E}\big[|\boldsymbol{X}_1 + \ldots + \boldsymbol{X}_t|^2\big] = \mathbb{E}\big[|\boldsymbol{X}_1|^2\big] + \ldots + \mathbb{E}\big[|\boldsymbol{X}_t|^2\big]. \tag{A.1}$$

Indeed, this type of expansion is used in Polyak and Juditsky [1992] to show $L^2$ convergence in the normality analysis of stochastic approximation problems.

However, correlatedness is *only* defined when random elements have *finite* variance. The following lemma provides an infinite-variance version of expansion (A.1), stating that the $p$-th moment ($p < 2$) of a martingale without square-integrability assumption can also be bounded *simpliciter* by the sum of the $p$-th moments of its differences, at the cost of a multiplicative constant that may depend only on $p$ and the dimension $n$. It is a generalization of the recent study Cherapanamjeri et al. [2020, Lemma 4.2].

**Lemma 7.** *Suppose $p \in [0, 1]$ and let $\{\boldsymbol{S}_t\}_{t\in\mathbb{N}}$ be an $n$-dimensional martingale adapted to the filtration $\{\mathcal{F}_t\}_{t\in\mathbb{N}}$, with $\mathbb{E}[|\boldsymbol{S}_t|^{1+p}] < \infty$ for every $t$ and $\boldsymbol{S}_0 = 0$. Let $\boldsymbol{X}_i = \boldsymbol{S}_i - \boldsymbol{S}_{i-1}$. Then*

$$\mathbb{E}\Big[|\boldsymbol{S}_t|^{1+p}\Big] \leqslant 2^{1-p}n^{1-\frac{1+p}{2}} \sum_{i=1}^{t} \mathbb{E}\Big[|\boldsymbol{X}_i|^{1+p}\Big].$$

Next, we present a Taylor-expansion-type inequality for the function $\|\boldsymbol{x}\|_p^p$. Recall that we have defined the signed power of a vector in (3.1).

**Lemma 8.** *Let $p \in [1, 2]$. For any $\boldsymbol{x}, \boldsymbol{y} \in \mathbb{R}^n$, $\|\boldsymbol{x} + \boldsymbol{y}\|_p^p \leqslant \|\boldsymbol{x}\|_p^p + 4\|\boldsymbol{y}\|_p^p + p\boldsymbol{y}^\mathsf{T}\boldsymbol{x}^{\langle p-1\rangle}$.*

This inequality traces back to Krasulina [1969], where the one-dimensional version $|x + y|^p \leqslant |x|^p + C|y|^p + pyx^{p-1}\operatorname{sign}(x)$ is used[3] to derive an $L^p$ rate of convergence for the one-dimensional stochastic approximation process with step-size $1/t$. In our current study, this lemma is used not only to derive $L^p$ rate of convergence for general infinite-variance process in $\mathbb{R}^n$ with variable step-size scheme (Theorem 3), but also in the proof of the equivalent definitions of $p$-PD (Theorem 10).

Finally, we quote Fabian [1967, Lemma 4.2], which we shall use to calculate the exact convergence rate (see also Chung [1954]).

**Lemma 9** (Fabian [1967], Lemma 4.2)**.** *Let $\{b_t\}_{t\in\mathbb{N}}, A, B, \alpha, \beta$ be real numbers such that $0 < \alpha < 1$, $A > 0$ and suppose the recursion*

$$b_{t+1} = b_t(1 - At^{-\alpha}) + Bt^{-\alpha-\beta}$$

*holds. Then, $b_t = \Theta(t^{-\beta})$.*

### A.2   Discussions on $p$-Positive Definiteness and Uniform $p$-Positive Definiteness

Let us now focus on $p$-PD and uniform $p$-PD conditions which are defined in Definition 1, Definition 2 (also see Assumption 1). The next theorem provides several equivalent characterizations of $p$-PD condition, which will be used in the proof of $L^p$ convergence.

---

[3]The paper Krasulina [1969] contains a minor error in ignoring the signum function $\operatorname{sign}(x)$ in this inequality. Our proof of Theorem 3 can be thought of its correction as well as extension.

**Theorem 10** (Equivalent definitions of $p$-PD). *Let $\mathbf{Q}$ be a symmetric matrix. The following are equivalent when $p \in [1, 2]$.*

    *i) There exist $\delta, L > 0$, such that $\|\mathbf{I} - t\mathbf{Q}\|_p^p \leqslant 1 - Lt$ for all $t \in [0, \delta)$.*

    *ii) There exists $\lambda > 0$ such that for all $\boldsymbol{v} \in \mathbb{R}^n$, $\boldsymbol{v}^\mathsf{T} \mathbf{Q} \boldsymbol{v}^{\langle p-1 \rangle} \geqslant \lambda \|\boldsymbol{v}\|_p^p$.*

    *iii) For all $\boldsymbol{v} \in S_p$, $\boldsymbol{v}^\mathsf{T} \mathbf{Q} \boldsymbol{v}^{\langle p-1 \rangle} > 0$.*

    *iv) For all $\boldsymbol{v} \in S_p$, there exists $t_0 > 0$ such that $\|\boldsymbol{v} - t_0 \mathbf{Q} \boldsymbol{v}\|_p < 1$.*

Next, we provide several equivalent characterizations of uniform $p$-PD.

**Theorem 11** (Equivalent definitions of uniform $p$-PD). *Let $\mathcal{M}$ be a bounded set of symmetric matrices. The following are equivalent when $p \in [1, 2]$.*

    *i) There exist $\delta, L > 0$, such that $\|\mathbf{I} - t\mathbf{Q}\|_p^p \leqslant 1 - Lt$ for all $t \in [0, \delta)$ and $\mathbf{Q} \in \mathcal{M}$.*

    *ii) There exists $\lambda > 0$ such that for all $\boldsymbol{v} \in \mathbb{R}^n$ and $\mathbf{Q} \in \mathcal{M}$, $\boldsymbol{v}^\mathsf{T} \mathbf{Q} \boldsymbol{v}^{\langle p-1 \rangle} \geqslant \lambda \|\boldsymbol{v}\|_p^p$.*

    *iii) For all $\boldsymbol{v} \in S_p$ and $\mathbf{Q} \in \overline{\mathcal{M}}$, $\boldsymbol{v}^\mathsf{T} \mathbf{Q} \boldsymbol{v}^{\langle p-1 \rangle} > 0$.*

    *iv) For all $\boldsymbol{v} \in S_p$ and $\mathbf{Q} \in \overline{\mathcal{M}}$, there exists $t_0 > 0$ such that $\|\boldsymbol{v} - t_0 \mathbf{Q} \boldsymbol{v}\|_p < 1$.*

We notice that some mild assumptions can indeed imply $p$-PD. For example, we will show that diagonal dominance implies $p$-PD. Recall that a symmetric matrix $\mathbf{Q} = (q_{ij})_{n \times n}$ is called diagonally dominant (with non-negative diagonal) if for every $i \in [n]$,

$$q_{ii} - \sum_{j \in [n] \setminus \{i\}} |q_{ij}| > 0.$$

Further, we say that a non-empty set $\mathcal{M}$ of symmetric matrices is *uniformly diagonally dominant* (with non-negative diagonal) if

$$\inf_{(q_{ij})_{n \times n} \in \mathcal{M}} \min_{i \in [n]} \left( q_{ii} - \sum_{j \in [n] \setminus \{i\}} |q_{ij}| \right) > 0.$$

We have the following observations which we shall prove in Section B. First, we observe that the uniform $p$-PD assumption is weaker than the notion of uniform diagonally dominance (with non-negative diagonal).

**Proposition 12.** *A uniformly diagonally dominant (with non-negative diagonal) set of symmetric matrices is uniformly $p$-PD for every $p \in [1, 2]$.*

Next, we notice that the result in Proposition 12 is tight for $p = 1$.

**Proposition 13.** *Uniform $1$-PD is equivalent to uniform diagonal dominance (with non-negative diagonal).*

Finally, we observe that the notion of uniform 2-PD is weaker than uniform $p$-PD for any $p \in [1, 2]$.

**Proposition 14.** *Let $p \in [1, 2]$. Uniform $p$-PD implies uniform 2-PD.*

## B   Omitted Proofs

In this section, we first prove the lemmas, theorems, and propositions in Section A, then prove the theorems in Sections 3 and 4. Throughout this section, we denote by $\boldsymbol{\delta}_t$ the error of the approximation $\boldsymbol{x}_t - \boldsymbol{x}^*$, and by $\overline{\boldsymbol{\delta}}_t$ the averaged error $(\boldsymbol{\delta}_0 + \ldots + \boldsymbol{\delta}_{t-1})/t$. The gradient $\boldsymbol{\nabla} f(\boldsymbol{x})$ and the Hessian $\boldsymbol{\nabla}^2 f(\boldsymbol{x})$ will be written as $\boldsymbol{R}(\boldsymbol{x})$ and $\boldsymbol{\nabla} \boldsymbol{R}(\boldsymbol{x})$ respectively, not only for notational simplicity, but also to stress the fact that our results can be applied to any instance of stochastic approximation (2.1) including SGD.

**Proof of Lemma 7** We first prove the $n = 1$ case. Suppose $\{S_t\}$ is a one-dimensional martingale and $X_i = S_i - S_{i-1}$. Notice that the function $g(x) = |x|^{1+p}$ satisfies the inequality (see e.g. Cherapanamjeri et al. [2020, Lemma A.3]):

$$|g'(x) - g'(y)| \leqslant 2^{1-p} g'(|x - y|),$$

where the weak derivative $g'(x) = \text{sign}(x)$ is used in the inequality above in the case of $p = 0$, where

$$\text{sign}(x) := \begin{cases} 1 & \text{if } x > 0, \\ -1 & \text{if } x < 0, \\ 0 & \text{if } x = 0. \end{cases}$$

Furthermore, by $\mathbb{E}[X_i g'(S_{i-1}) \mid \mathcal{F}_{i-1}] = g'(S_{i-1})\mathbb{E}[X_i \mid \mathcal{F}_{i-1}] = 0$, we have

$$
\begin{aligned}
\mathbb{E}[g(S_t)] &= \sum_{i=1}^{t} \mathbb{E}\left[\int_{S_{i-1}}^{S_i} g'(x)\mathrm{d}x\right] \\
&= \sum_{i=1}^{t} \mathbb{E}\left[X_i g'(S_{i-1}) + \int_{S_{i-1}}^{S_i} [g'(x) - g'(S_{i-1})]\mathrm{d}x\right] \\
&= \sum_{i=1}^{t} \mathbb{E}\left[\int_{S_{i-1}}^{S_i} [g'(x) - g'(S_{i-1})]\mathrm{d}x\right] \\
&= \sum_{i=1}^{t} \mathbb{E}\left[\int_{0}^{X_i} [g'(S_{i-1} + \tau) - g'(S_{i-1})]\mathrm{d}\tau\right] \\
&= \sum_{i=1}^{t} \mathbb{E}\left[\int_{0}^{|X_i|} |g'(S_{i-1} + \text{sign}(X_i)\tau) - g'(S_{i-1})|\mathrm{d}\tau\right] \\
&\leqslant 2^{1-p} \sum_{i=1}^{t} \mathbb{E}\left[\int_{0}^{|X_i|} g'(\tau)\mathrm{d}\tau\right] \\
&= 2^{1-p} \sum_{i=1}^{t} \mathbb{E}[g(|X_i|)].
\end{aligned}
\tag{B.1}
$$

Next, for the higher dimension $n > 1$, we denote by $S_i^j$ (resp. $X_i^j$) the $j$-th entry of the vector $\boldsymbol{S}_i$ (resp. $\boldsymbol{X}_i$). We can apply the inequality (B.1) obtained above to $S_t^j$ by taking a $(1+p)$-norm,

$$
\begin{aligned}
\mathbb{E}\left[\|\boldsymbol{S}_t\|_{1+p}^{1+p}\right] &= \sum_{j=1}^{n} \mathbb{E}\left[\left|S_t^j\right|^{1+p}\right] \\
&\leqslant \sum_{j=1}^{n} 2^{1-p} \sum_{i=1}^{t} \mathbb{E}\left[\left|X_i^j\right|^{1+p}\right] \\
&= 2^{1-p} \sum_{i=1}^{t} \sum_{j=1}^{n} \mathbb{E}\left[\left|X_i^j\right|^{1+p}\right] \\
&= 2^{1-p} \sum_{i=1}^{t} \mathbb{E}\left[\|\boldsymbol{X}_i\|_{1+p}^{1+p}\right].
\end{aligned}
$$

Finally, the inequalities

$$|\boldsymbol{x}| \leqslant \|\boldsymbol{x}\|_{1+p} \leqslant n^{\frac{1}{1+p} - \frac{1}{2}} |\boldsymbol{x}|$$

give our desired result:

$$\mathbb{E}\left[|\boldsymbol{S}_t|^{1+p}\right] \leqslant 2^{1-p} n^{1 - \frac{1+p}{2}} \sum_{i=1}^{t} \mathbb{E}\left[|\boldsymbol{X}_i|^{1+p}\right].$$

The proof is complete. □

**Proof of Lemma** 8    By the inequality that $|1 + a|^p \leqslant 1 + ap + 4|a|^p$ for any $p \in [1, 2]$ and $a \in \mathbb{R}$, we have that for any $p \in [1, 2]$ and $x, y \in \mathbb{R}$,

$$|x + y|^p \leqslant |x|^p + py|x|^{p-1} \operatorname{sign}(x) + 4|y|^p. \tag{B.2}$$

Next, for any $\boldsymbol{x} = (x^1, \ldots, x^n)^\mathsf{T}, \boldsymbol{y} = (y^1, \ldots, y^n)^\mathsf{T} \in \mathbb{R}^n$, by taking the $p$-norm and applying the inequality (B.2), we obtain

$$
\begin{aligned}
\|\boldsymbol{x} + \boldsymbol{y}\|_p^p &= \sum_{i=1}^n |x^i + y^i|^p \\
&\leqslant \sum_{i=1}^n \left( |x^i|^p + py^i|x^i|^{p-1} \operatorname{sign}(x^i) + 4|y^i|^p \right) \\
&= \|\boldsymbol{x}\|_p^p + 4\|\boldsymbol{y}\|_p^p + p\sum_{i=1}^n y^i|x^i|^{p-1} \operatorname{sign}(x^i) \\
&= \|\boldsymbol{x}\|_p^p + 4\|\boldsymbol{y}\|_p^p + p\boldsymbol{y}^\mathsf{T}\boldsymbol{x}^{\langle p-1 \rangle},
\end{aligned}
$$

which completes the proof. □

Since Theorem 10 is just a special case of Theorem 11, we will only prove the latter. Before we proceed, let us first state a useful technical lemma.

**Lemma 15.** *Let $\boldsymbol{u}, \boldsymbol{v} \in \mathbb{R}^n$ and consider the function $\varphi(t) = \|\boldsymbol{u} + t\boldsymbol{v}\|_p^p = \sum_{i=1}^n |u^i + v^i t|^p$. The function $\varphi$ is convex and has the following derivative (when $1 < p \leqslant 2$) or subderivative (when $p = 1$):*

$$\varphi'(t) = \sum_{i=1}^n p|u^i + v^i t|^{p-1} \operatorname{sign}(u^i + v^i t)v^i = p\boldsymbol{v}^\mathsf{T}(\boldsymbol{u} + t\boldsymbol{v})^{\langle p-1 \rangle}.$$

The proof of Lemma 15 is straightforward and is hence omitted here.

Now we are ready to prove Theorem 11.

**Proof of Theorem** 11    We shall show that i) $\implies$ iv) $\implies$ iii) $\implies$ ii) $\implies$ i).

- i) $\implies$ iv) Take a sequence $\{\mathbf{Q}_1, \mathbf{Q}_2, \ldots\} \subseteq \mathcal{M}$ such that $\lim_{m \to \infty} \mathbf{Q}_m = \mathbf{Q}$. iv) follows from $\|\mathbf{I} - (\delta/2)\mathbf{Q}_m\|_p^p \leqslant 1 - L\delta/2$.

- iv) $\implies$ iii) For all $\boldsymbol{v} \in S_p$ and $\mathbf{Q} \in \overline{\mathcal{M}}$, consider the function $\varphi(t) = \|\boldsymbol{v} - t\mathbf{Q}\boldsymbol{v}\|_p^p$. According to Lemma 15, $\varphi(t)$ is convex. Furthermore, $\varphi(t_0) < 1 = \varphi(0)$. Hence it follows that $\varphi'(0) < 0$; that is, $\boldsymbol{v}^\mathsf{T}\mathbf{Q}\boldsymbol{v}^{\langle p-1 \rangle} > 0$.

- iii) $\implies$ ii) Since the function $(\boldsymbol{v}, \mathbf{Q}) \mapsto \boldsymbol{v}^\mathsf{T}\mathbf{Q}\boldsymbol{v}^{\langle p-1 \rangle}$ is continuous, it maps the compact set $S_p \times \overline{\mathcal{M}}$ to a compact set. Hence there exists some $\lambda > 0$ such that for all $\boldsymbol{v} \in S_p$ and $\mathbf{Q} \in \overline{\mathcal{M}}$, $\boldsymbol{v}^\mathsf{T}\mathbf{Q}\boldsymbol{v}^{\langle p-1 \rangle} \geqslant \lambda$. Now, for every $\boldsymbol{u} \in \mathbb{R}^n \setminus \{0\}$, by setting $\boldsymbol{v} = \boldsymbol{u}/\|\boldsymbol{u}\|_p$, we get $\boldsymbol{u}^\mathsf{T}\mathbf{Q}\boldsymbol{u}^{\langle p-1 \rangle} \geqslant \lambda\|\boldsymbol{u}\|_p^p$.

- ii) $\implies$ i) For arbitrary $\boldsymbol{v} \in \mathbb{R}^n$ and $\mathbf{Q} \in \mathcal{M}$, by Lemma 8 we have $\|(\mathbf{I} - t\mathbf{Q})\boldsymbol{v}\|_p^p = \|\boldsymbol{v} - t\mathbf{Q}\boldsymbol{v}\|_p^p \leqslant \|\boldsymbol{v}\|_p^p + 4t^p\|\mathbf{Q}\boldsymbol{v}\|_p^p - pt(\boldsymbol{v}^\mathsf{T}\mathbf{Q}\boldsymbol{v}^{\langle p-1 \rangle}) \leqslant \|\boldsymbol{v}\|_p^p + 4t^p\|\mathbf{Q}\|_p^p\|\boldsymbol{v}\|_p^p - pt\lambda\|\boldsymbol{v}\|_p^p$. This implies i).

The proof is complete. □

**Proof of Proposition** 12     Let $\mathbf{Q} \in \mathcal{M}$ and $\boldsymbol{v} \in \mathbb{R}^n$.

$$\boldsymbol{v}^\top \mathbf{Q} \boldsymbol{v}^{\langle p-1 \rangle} = \sum_{i=1}^n q_{ii} |v^i|^p + \sum_{i<j} q_{ij}(v^i |v^j|^{p-1} \operatorname{sign}(v^j) + v^j |v^i|^{p-1} \operatorname{sign}(v^i))$$

$$\geqslant \sum_{i=1}^n q_{ii} |v^i|^p - \sum_{i<j} |q_{ij}|(|v^i||v^j|^{p-1} + |v^j||v^i|^{p-1})$$

$$\geqslant \sum_{i=1}^n q_{ii} |v^i|^p - \sum_{i<j} |q_{ij}|(|v^i|^p + |v^j|^p)$$

$$= \sum_{i=1}^n |v^i|^p \left( q_{ii} - \sum_{j \neq i} |q_{ij}| \right),$$

where we used the inequality $x^p + y^p \geqslant x^{p-1}y + y^{p-1}x$ for any $p \geqslant 1$ and $x, y \geqslant 0$[4] to get the third line from the second line above. Hence the uniform $p$-PD of $\mathcal{M}$ follows from the item ii) of Theorem 11. The proof is complete.                                    □

**Proof of Proposition** 13     Suppose $\mathcal{M}$ is uniform 1-PD. By the item i) of Theorem 11, there exists $\delta, L > 0$ such that $\|\mathbf{I} - t\mathbf{Q}\|_1 \leqslant 1 - Lt$ for all $t \in [0, \delta)$ and $\mathbf{Q} \in \mathcal{M}$. Let $\mathbf{Q} = (q_{ij})_{n \times n}$ and notice that

$$\|\mathbf{I} - t\mathbf{Q}\|_1 = \max_{i \in [n]} \left( |1 - tq_{ii}| + \sum_{j \in [n] \setminus \{i\}} t|q_{ij}| \right).$$

It follows that

$$\min_{i \in [n]} \left( q_{ii} - \sum_{j \in [n] \setminus \{i\}} |q_{ij}| \right) \geqslant L > 0.$$

Hence $\mathcal{M}$ is uniformly diagonally dominant (with non-negative diagonal). The proof is complete.     □

**Proof of Proposition** 14     Suppose $\mathcal{M}$ is uniformly $p$-PD but not uniformly 2-PD. Then, there exists a sequence $\{\mathbf{Q}_1, \mathbf{Q}_2, \ldots\} \subseteq \mathcal{M}$ such that the smallest eigenvalues $\lambda_m$ of $\mathbf{Q}_m$ satisfy

$$\lim_{m \to \infty} \lambda_m \leqslant 0. \tag{B.3}$$

For each $m \in \mathbb{N}^+$, there exists an $\boldsymbol{v}_m \in \mathbb{R}^n \setminus \{0\}$ such that $\mathbf{Q}_m \boldsymbol{v}_m = \lambda_m \boldsymbol{v}_m$. Hence

$$\boldsymbol{v}_m^\top \mathbf{Q}_m \boldsymbol{v}_m^{\langle p-1 \rangle} = \lambda_m \boldsymbol{v}_m^\top \boldsymbol{v}_m^{\langle p-1 \rangle} = \lambda_m \|\boldsymbol{v}_m\|_p^p.$$

But by the item ii) of Theorem 11, there exists $\lambda > 0$ such that $\lambda_m \geqslant \lambda$. This contradicts (B.3). The proof is complete.                                    □

**Proof of Theorem** 3     We use a technique similar to Krasulina [1969]. Define the function

$$\boldsymbol{T}_t(\boldsymbol{x}) = \left( T_t^1(\boldsymbol{x}), \ldots, T_t^n(\boldsymbol{x}) \right)^\top = \boldsymbol{x} - \boldsymbol{x}^* - \gamma_{t+1} \boldsymbol{R}(x).$$

An $n$-dimensional (and corrected) version of the first inequality in the proof of Krasulina [1969, Theorem 2] can be obtained by applying Lemma 8 to our stochastic approximation scheme,

$$\|\boldsymbol{x}_{t+1} - \boldsymbol{x}^*\|_p^p = \left\| \boldsymbol{T}_t(\boldsymbol{x}_t) - \gamma_{t+1} \boldsymbol{\xi}_{t+1} \right\|_p^p$$

$$\leqslant \|\boldsymbol{T}_t(\boldsymbol{x}_t)\|_p^p + 4\gamma_{t+1}^p \|\boldsymbol{\xi}_{t+1}\|_p^p + p\gamma_{t+1} \sum_{i=1}^n \xi_{t+1}^i \left| T_t^i(\boldsymbol{x}_t) \right|^{p-1} \operatorname{sign} T_t^i(\boldsymbol{x}_t) \tag{B.4}$$

Since $\mathbb{E}\left[ \xi_{t+1}^i |T_t^i(\boldsymbol{x}_t)|^{p-1} \operatorname{sign} T_t^i(\boldsymbol{x}_t) \mid \boldsymbol{x}_t \right] = |T_t^i(\boldsymbol{x}_t)|^{p-1} \operatorname{sign} T_t^i(\boldsymbol{x}_t) \mathbb{E}[\xi_{t+1}^i \mid \boldsymbol{x}_t] = 0$, by taking expectations in (B.4), we get

$$\mathbb{E}\left[ \|\boldsymbol{\delta}_{t+1}\|_p^p \right] = \mathbb{E}\left[ \|\boldsymbol{x}_{t+1} - \boldsymbol{x}^*\|_p^p \right]$$

$$\leqslant \mathbb{E}\left[ \|\boldsymbol{T}_t(\boldsymbol{x}_t)\|_p^p \right] + 4\gamma_{t+1}^p \mathbb{E}\left[ \|\boldsymbol{\xi}_{t+1}\|_p^p \right]$$

$$= \mathbb{E}\left[ \|(\boldsymbol{x}_t - \boldsymbol{x}^*) - \gamma_{t+1} \boldsymbol{R}(\boldsymbol{x}_t)\|_p^p \right] + 4\gamma_{t+1}^p \mathbb{E}\left[ \|\boldsymbol{\xi}_{t+1}\|_p^p \right].$$

---

[4]To see this, notice that for any $p \geqslant 1$ and $x, y \geqslant 0$, $x^p + y^p - x^{p-1}y - y^{p-1}x = (x^{p-1} - y^{p-1})(x-y) \geqslant 0$.

By the mean value theorem, there exists $x_t^\flat \in \{x^* + \tau(x_t - x^*) : 0 \leqslant \tau \leqslant 1\}$, such that $R(x_t) = \nabla R(x_t^\flat)(x_t - x^*)$, and then

$$\mathbb{E}\Big[\|(x_t - x^*) - \gamma_{t+1}R(x_t)\|_p^p\Big] + 4\gamma_{t+1}^p\mathbb{E}\Big[\|\xi_{t+1}\|_p^p\Big]$$

$$= \mathbb{E}\Big[\big\|(I - \gamma_{t+1}\nabla R(x_t^\flat))(x_t - x^*)\big\|_p^p\Big] + 4\gamma_{t+1}^p\mathbb{E}\Big[\|\xi_{t+1}\|_p^p\Big]$$

$$\leqslant \mathbb{E}\Big[\big\|I - \gamma_{t+1}\nabla R(x_t^\flat)\big\|_p^p \cdot \|x_t - x^*\|_p^p\Big] + 4\gamma_{t+1}^p\mathbb{E}\Big[\|\xi_{t+1}\|_p^p\Big]$$

$$\leqslant \mathbb{E}\Big[\big\|I - \gamma_{t+1}\nabla R(x_t^\flat)\big\|_p^p \cdot \|\delta_t\|_p^p\Big] + C_0\gamma_{t+1}^p\big(1 + \mathbb{E}\big[\|\delta_t\|_p^p\big]\big),$$

where the last inequality follows from

$$\mathbb{E}[|m_{t+1}|^p \mid \mathcal{F}_t] \leqslant \mathbb{E}\Big[|m_{t+1}|^2 \mid \mathcal{F}_t\Big]^{p/2} \leqslant \big[K\big(1 + |x_t|^2\big)\big]^{p/2} \tag{B.5}$$

$$\leqslant K^{p/2}(1 + |x_t|^p) \leqslant K^{p/2}\big(1 + 2^{p-1}(|\delta_t|^p + |x^*|^p)\big),$$

where we used the inequality $(x + y)^r \leqslant x^r + y^r$ for any $x, y \geqslant 0, 0 \leqslant r \leqslant 1$ to obtain the first inequality in the second line above, as well as the assumption $\mathbb{E}[|\zeta_1|^p] < \infty$.

Note that $\big\|I - \gamma_{t+1}\nabla R(x_t^\flat)\big\|_p^p$ can be estimated by the uniform $p$-PD assumption (see item i) of Theorem 11) since $\gamma_t \to 0$. For $t$ sufficiently large,

$$\big\|I - \gamma_{t+1}\nabla R(x_t^\flat)\big\|_p^p \leqslant 1 - L\gamma_{t+1}.$$

And there is a positive constant $C_1$ such that $1 - L\gamma_{t+1} + C_0\gamma_{t+1}^p \leqslant 1 - C_1\gamma_{t+1}$ for $t$ sufficiently large. Hence, we arrive at the following iterative bound

$$\mathbb{E}\Big[\|\delta_{t+1}\|_p^p\Big] \leqslant (1 - \gamma_{t+1}C_1) \cdot \mathbb{E}\Big[\|\delta_t\|_p^p\Big] + C_0\gamma_{t+1}^p \tag{B.6}$$

for $t$ sufficiently large.

Next, let us substitute $\gamma_{t+1}$ with $t^{-\rho}$ where $0 < \rho < 1$. Consider the iteration

$$\mu_{t+1} = (1 - t^{-\rho}C_1) \cdot \mu_t + C_0 t^{-\rho p}, \tag{B.7}$$

so that by (B.6), $\mathbb{E}\Big[\|\delta_t\|_p^p\Big] = \mathcal{O}(\mu_t)$. By virtue of Lemma 9, we get

$$\mu_t = \Theta\Big(t^{-\rho(p-1)}\Big). \tag{B.8}$$

Therefore, by (B.6), (B.7), and (B.8), we obtain the following rate of convergence:

$$\mathbb{E}[\|\delta_t\|_p^p] = \mathcal{O}\Big(t^{-\rho(p-1)}\Big).$$

Next, since $p$-norms on $\mathbb{R}^n$ are all equivalent, we can drop the subscript $\|\cdot\|_p$ and obtain

$$\mathbb{E}[|\delta_t|^p] = \mathcal{O}\Big(t^{-\rho(p-1)}\Big).$$

Finally, by (B.5), we see that $\sup_{t\in\mathbb{N}^+}\mathbb{E}[|\xi_t|^p] \leqslant \sup_{t\in\mathbb{N}^+}\mathbb{E}[2^{p-1}(|m_t|^p + |\zeta_t|^p)] < \infty$. The proof is complete. $\quad\square$ **Proof of Corollary** 4      Under the assumptions of Corollary 4, the rate $\mathbb{E}[|\delta_t|^p] = \mathcal{O}\big(t^{-\rho(p-1)}\big)$ holds for every $p \in [q, \alpha)$. We can thus apply Jensen's inequality to strengthen it. By Jensen's inequality and (3.4), we get

$$\mathbb{E}[|\delta_t|^q] \leqslant \mathbb{E}[|\delta_t|^p]^{q/p} = \mathcal{O}\Big(t^{-\rho(p-1)\frac{q}{p}}\Big).$$

By letting $p \nearrow \alpha$, we conclude that have for every $\varepsilon > 0$,

$$\mathbb{E}[|\delta_t|^q] = o\Big(t^{-\rho q\frac{\alpha-1}{\alpha}+\varepsilon}\Big).$$

The proof is complete. $\quad\square$

Next, we state a series of technical lemmas as well as their proofs, which will be used in the proofs of Theorems 5 and 6.

**Lemma 16.** *If $\gamma_t \asymp t^{-\rho}$ with $0 < \rho < \kappa \leqslant 1$, then for all $\lambda > 0$,*

$$\lim_{t\to\infty} t^{-\kappa} \sum_{j=1}^{t-1} \exp\left(-\lambda \sum_{i=j}^{t-1} \gamma_i\right) = 0.$$

**Proof.** Notice that there exists some constant $B > 0$ such that

$$\sum_{i=j}^{t-1} \gamma_i \geqslant \frac{B}{\lambda}\left(t^{1-\rho} - j^{1-\rho}\right).$$

It follows that

$$t^{-\kappa} \sum_{j=1}^{t-1} \exp\left(-\lambda \sum_{i=j}^{t-1} \gamma_i\right) \leqslant t^{-\kappa} \sum_{j=0}^{t-1} \exp\left(-Bt^{1-\rho} + Bj^{1-\rho}\right) = \frac{\sum_{j=0}^{t-1} \exp(Bj^{1-\rho})}{t^\kappa \exp(Bt^{1-\rho})}.$$

By Stolz-Cesàro theorem, we have

$$\frac{\sum_{j=0}^{t-1} \exp(Bj^{1-\rho})}{t^\kappa \exp(Bt^{1-\rho})} \asymp \frac{\exp(Bt^{1-\rho})}{(t+1)^\kappa \exp(B(t+1)^{1-\rho}) - t^\kappa \exp(Bt^{1-\rho})}$$

$$= \frac{1}{(t+1)^\kappa \exp[B((t+1)^{1-\rho} - t^{1-\rho})] - t^\kappa}$$

$$= \frac{1}{(t+1)^\kappa \exp[B(1-\rho)(t+1)^{-\rho} + o(t^{-\rho})] - t^\kappa}$$

$$= \frac{1}{(t+1)^\kappa [1 + B(1-\rho)(t+1)^{-\rho} + o(t^{-\rho})] - t^\kappa}$$

$$= \frac{1}{B(1-\rho)(t+1)^{\kappa-\rho} + o((t+1)^{\kappa-\rho})}$$

$$\to 0,$$

as $t \to \infty$. The proof is complete. $\qquad\square$

**Lemma 17.** *Suppose $\gamma_t \asymp t^{-\rho}$ and $0 < \rho < \kappa \leqslant 1$; let $\mathbf{A}$ be a positive definite symmetric matrix. Consider the matrix recursion in [Polyak and Juditsky, 1992, Lemma 1],*

$$\mathbf{X}_j^j = \mathbf{I}, \quad \mathbf{X}_j^{t+1} = \mathbf{X}_j^t - \gamma_t \mathbf{A} \mathbf{X}_j^t, \quad (j \in \mathbb{N}^+)$$

*and define*

$$\overline{\mathbf{X}}_j^t = \gamma_j \sum_{i=j}^{t-1} \mathbf{X}_j^i, \quad \mathbf{\Phi}_j^t = \mathbf{A}^{-1} - \overline{\mathbf{X}}_j^t.$$

*Then the following limit holds,*

$$\lim_{t\to\infty} \frac{1}{t^\kappa} \sum_{j=1}^{t-1} \|\mathbf{\Phi}_j^t\| = 0.$$

**Remark.** Lemma 17 recovers [Polyak and Juditsky, 1992, Lemma 1] as the special case $\kappa = 1$.

**Proof of Lemma 17** Modeling after Polyak and Juditsky [1992]'s proof of their Lemma 1, we define $\mathbf{S}_j^t = \sum_{i=j}^{t-1}(\gamma_i - \gamma_j)\mathbf{X}_j^i$, and we have

$$\mathbf{\Phi}_j^t = \mathbf{S}_j^t + \mathbf{A}^{-1}\mathbf{X}_j^t.$$

We will split the proofs into two parts. In the first part, we will prove $t^{-\kappa} \sum_{j=1}^{t-1} \|\mathbf{S}_j^t\| \to 0$ and then in the second part we will prove $t^{-\kappa} \sum_{j=1}^{t-1} \|\mathbf{X}_j^t\| \to 0$.

**Part I.** We first prove that $t^{-\kappa} \sum_{j=1}^{t-1} \|\mathbf{S}_j^t\| \to 0$.

By the Part 3 of Polyak and Juditsky [1992, Lemma 1][5], there exist some $\lambda > 0$ and $K < \infty$ such that

$$\|\mathbf{X}_j^t\| \leqslant K \exp\left(-2\lambda \sum_{i=j}^{t-1} \gamma_i\right) = Ke^{-2\lambda m_j^t}, \tag{B.9}$$

where $m_k^\ell$ stands for $\sum_{i=k}^{\ell-1} \gamma_i$. Now we have

$$\begin{aligned}
\|\mathbf{S}_j^t\| &= \left\| \sum_{i=1}^t (\gamma_i - \gamma_j) \mathbf{X}_j^i \right\| \\
&= \left\| \sum_{i=1}^t \left[ \sum_{k=j}^{i-1} (\gamma_{k+1} - \gamma_k) \right] \mathbf{X}_j^i \right\| \\
&\leqslant C_0 \sum_{i=j}^t \sum_{k=j}^{i-1} k^{-\rho-1} \exp\left(-2\lambda m_j^i\right) \\
&\leqslant C_0 j^{-1} \sum_{i=j}^t \sum_{k=j}^{i-1} k^{-\rho} \exp\left(-2\lambda m_j^i\right) \\
&\leqslant C_1 j^{-1} \sum_{i=j}^t m_j^i \exp\left(-2\lambda m_j^i\right) \\
&= C_1 j^{-1} \sum_{i=j}^t \frac{m_j^i e^{-2\lambda m_j^i} (m_j^i - m_j^{i-1})}{\gamma_i}, \tag{B.10}
\end{aligned}$$

where $C_0, C_1$ are some positive constants.

Since the function $f_w(x) = x^\rho \exp(-wx^{1-\rho})$ is bounded on $x \in [1, \infty)$ for every $w > 0$, we have

$$\frac{j^{-\rho}}{\gamma_i} \exp\left(-\lambda m_j^i\right) \leqslant C_2 i^\rho j^{-\rho} \exp(-C_3(i^{1-\rho} - j^{1-\rho})) = C_2 f_{C_3}(i)/f_{C_3}(j) \leqslant C_4,$$

for some positive constants $C_2$, $C_3$ and $C_4$. Hence, continuing upon (B.10),

$$\|\mathbf{S}_j^t\| \leqslant C_1 C_4 j^{\rho-1} \sum_{i=j}^t m_j^i e^{-\lambda m_j^i} (m_j^i - m_j^{i-1}).$$

Since the summation $\sum_{i=j}^t m_j^i e^{-\lambda m_j^i} (m_j^i - m_j^{i-1})$ approximates $\int_0^{m_j^t} m e^{-\lambda m} \mathrm{d}m$, it is bounded. Hence, for some positive constant $C_5$,

$$\|\mathbf{S}_j^t\| \leqslant C_5 j^{\rho-1},$$

which implies the desired limit

$$\lim_{t\to\infty} t^{-\kappa} \sum_{j=1}^{t-1} \|\mathbf{S}_j^t\| = 0.$$

**Part II.** It remains to prove that $t^{-\kappa} \sum_{j=1}^{t-1} \|\mathbf{X}_j^t\| \to 0$.

Combining (B.9) and Lemma 16, we have $t^{-\kappa} \sum_{j=1}^{t-1} \|\mathbf{X}_j^t\| \to 0$. Hence the proof of this lemma is complete. $\qquad\square$

**Lemma 18.** *Given the assumption of Theorem 5 or Theorem 6,*

$$\frac{\boldsymbol{\xi}_1 + \cdots \boldsymbol{\xi}_t}{t^{1/\alpha}} \xrightarrow[t\to\infty]{\mathcal{D}} \mu.$$

---

[5]We can directly use this inequality since our assumption on step-size $\gamma_t \asymp t^{-\rho}, 0 < \rho < 1$ can meet Polyak and Juditsky [1992, Assumption 2.2].

**Proof.** We recall the decomposition $\boldsymbol{\xi}_t = \boldsymbol{\zeta}_t + \boldsymbol{m}_t$, where $\{\boldsymbol{\zeta}_t\}$ are i.i.d. and $\boldsymbol{\zeta}_1$ is in the domain of normal attraction of an $n$-dimensional centered $\alpha$-stable distribution so that

$$\frac{\boldsymbol{\zeta}_1 + \ldots + \boldsymbol{\zeta}_t}{t^{1/\alpha}} \xrightarrow[t\to\infty]{\mathcal{D}} \mu.$$

Hence, it suffices to show that $t^{-1/\alpha}(\boldsymbol{m}_1 + \ldots + \boldsymbol{m}_t) \to 0$ in $L^r$, for some $r \geqslant 1$.

By (3.3), there exists a constant $C > 0$ such that

$$\mathbb{E}\Big[|\boldsymbol{m}_{t+1}(\boldsymbol{x}_t)|^2 \mid \mathcal{F}_t\Big] \leqslant K\big(1 + |\boldsymbol{x}_t|^2\big) \leqslant K(1 + 2|\boldsymbol{x}^*|^2 + 2|\boldsymbol{\delta}_t|^2) \leqslant C(1 + |\boldsymbol{\delta}_t|^2).$$

Hence, by using the "Remark" on p.151 of Neveu [1975] (cf. inequalities (20) of Anantharam and Borkar [2012]), we get

$$
\begin{aligned}
\mathbb{E}\left[\left|\frac{\boldsymbol{m}_1 + \ldots + \boldsymbol{m}_t}{t^{1/\alpha}}\right|^r\right] &\leqslant \frac{C_1}{t^{r/\alpha}} \mathbb{E}\left[\left(\sum_{i=1}^t \mathbb{E}\big[|\boldsymbol{m}_i|^2 \mid \mathcal{F}_{i-1}\big]\right)^{r/2}\right] \\
&\leqslant \frac{C_2}{t^{r/\alpha}} \mathbb{E}\left[\left(\sum_{i=1}^t \big(1 + |\boldsymbol{\delta}_{i-1}|^2\big)\right)^{r/2}\right] \\
&\leqslant \frac{C_2}{t^{r/\alpha}} \mathbb{E}\left[t^{r/2} + \sum_{i=1}^t |\boldsymbol{\delta}_{i-1}|^r\right],
\end{aligned}
\tag{B.11}
$$

where, for the last inequality, we use the fact that $(x + y)^s \leqslant x^s + y^s$ for any $x, y \geqslant 0$, $0 \leqslant s \leqslant 1$. If the assumption of Theorem 5 holds, take $r = p > (\alpha + \alpha\rho)/(1 + \alpha\rho)$ in the inequalities (B.11) above. Then, by Theorem 3, $\mathbb{E}[|\boldsymbol{\delta}_t|^r] = \mathcal{O}(t^{-\rho(r-1)}) = o(t^{r/\alpha-1})$.

If the assumption of Theorem 6 holds, take $r = q > 1/\rho > \alpha/(1 + \rho(\alpha - 1))$ in the inequalities (B.11) above. Then by Corollary 4, $\mathbb{E}[|\boldsymbol{\delta}_t|^r] = \tilde{\mathcal{O}}(t^{-\rho r(\alpha-1)/\alpha}) = o(t^{r/\alpha-1})$.

In both cases, $t^{-1/\alpha}(\boldsymbol{m}_1 + \ldots + \boldsymbol{m}_t) \to 0$ in $L^r$. The proof is complete. $\qquad\square$

Finally, we are ready to prove Theorems 5 and 6.

**Proof of Theorem** 5  By Polyak and Juditsky [1992, Lemma 2]:

$$\frac{t}{t^{1/\alpha}}\overline{\boldsymbol{\delta}}_t = \underbrace{\frac{1}{t^{1/\alpha}}\mathbf{F}_t\boldsymbol{\delta}_0}_{\boldsymbol{I}_t^{(1)}} - \underbrace{\frac{1}{t^{1/\alpha}}\sum_{j=1}^{t-1}\mathbf{A}^{-1}\boldsymbol{\xi}_j}_{\boldsymbol{I}_t^{(2)}} - \underbrace{\frac{1}{t^{1/\alpha}}\sum_{j=1}^{t-1}\mathbf{W}_j^t\boldsymbol{\xi}_j}_{\boldsymbol{I}_t^{(3)}}, \tag{B.12}$$

where $\mathbf{F}_t$ and $\mathbf{W}_j^t$ are deterministic matrices with uniformly bounded operator 2-norms defined as

$$\mathbf{F}_t = \sum_{i=0}^{t-1}\prod_{k=1}^i(\mathbf{I} - \gamma_k\mathbf{A}), \tag{B.13}$$

$$\mathbf{W}_j^t = \gamma_j\sum_{i=j}^{t-1}\prod_{k=j+1}^i(\mathbf{I} - \gamma_k\mathbf{A}) - \mathbf{A}^{-1}. \tag{B.14}$$

We have $\boldsymbol{I}_t^{(1)} \to 0$ by the boundedness of $\mathbf{F}_t$. Next, take some $\kappa$ such that

$$\max(\rho, 1/\alpha) < \kappa \leqslant p/\alpha. \tag{B.15}$$

We shall prove that $\boldsymbol{I}_t^{(3)} \to 0$ in $L^{\alpha\kappa}$ (notice that $1 < \alpha\kappa \leqslant p < \alpha$; cf. Polyak and Juditsky [1992, Proof of Theorem 1] where convergence in $L^2$ is proven). By Theorem 3, $\sup_j \mathbb{E}[|\boldsymbol{\xi}_j|^p] < \infty$. Hence

we can compute, by virtue of Lemma 7, that

$$\mathbb{E}\Big[\big|\boldsymbol{I}_t^{(3)}\big|^{\alpha\kappa}\Big] = \mathbb{E}\Bigg[\bigg|\frac{1}{t^{1/\alpha}}\sum_{j=1}^{t-1}\mathbf{W}_j^t\boldsymbol{\xi}_j\bigg|^{\alpha\kappa}\Bigg] \leqslant \frac{C_0}{t^\kappa}\sum_{j=1}^{t-1}\mathbb{E}\big[|\mathbf{W}_j^t\boldsymbol{\xi}_j|^{\alpha\kappa}\big]$$

$$\leqslant \Bigg(\frac{C_0}{t^\kappa}\sum_{j=1}^{t-1}\|\mathbf{W}_j^t\|^{\alpha\kappa}\Bigg)\sup_j\mathbb{E}\big[|\boldsymbol{\xi}_j|^{\alpha\kappa}\big] \leqslant \Bigg(\frac{C_0}{t^\kappa}\sum_{j=1}^{t-1}\|\mathbf{W}_j^t\|\Bigg)\sup_j\mathbb{E}\big[|\boldsymbol{\xi}_j|^{\alpha\kappa}\big]$$

$$\leqslant \frac{C_1}{t^\kappa}\sum_{j=1}^{t-1}\|\mathbf{W}_j^t\|.$$

Notice that the matrices $\mathbf{W}_j^t$ defined above correspond to $-\boldsymbol{\Phi}_j^t$ in Lemma 17. This infers that $\mathbb{E}\Big[|\boldsymbol{I}_t^{(3)}|^{\alpha\kappa}\Big] \leqslant \frac{K_1}{t^\kappa}\sum_{j=1}^{t-1}\|\mathbf{W}_j^t\| \to 0$ as $t \to \infty$.

Finally, Lemma 18 states that $\boldsymbol{I}_t^{(2)}$ converges weakly to an $\alpha$-stable distribution. Hence we conclude the proof. $\qquad\square$

**Proof of Theorem** 6    Denote by $\mathbf{A}$ the Hessian matrix $\nabla R(\boldsymbol{x}^*) = \nabla^2 f(\boldsymbol{x}^*)$. Consider a corresponding linear SA process with the same noise,

$$\boldsymbol{x}_{t+1}^1 = \boldsymbol{x}_t^1 - \gamma_{t+1}\big(\mathbf{A}(\boldsymbol{x}_t^1 - \boldsymbol{x}^*) + \boldsymbol{\xi}_{t+1}(\boldsymbol{x}_t)\big), \tag{B.16}$$

with $\boldsymbol{x}_0^1 = \boldsymbol{x}_0$. We further define $\boldsymbol{\delta}_t^1 = \boldsymbol{x}_t^1 - \boldsymbol{x}^*$ and the averaging process $\overline{\boldsymbol{\delta}}_t^1 = (\boldsymbol{\delta}_0^1 + \ldots + \boldsymbol{\delta}_{t-1}^1)/t$.

**Part I.** We first prove that $t^{1-1/\alpha}\big(\overline{\boldsymbol{\delta}}_t^1 - \overline{\boldsymbol{\delta}}_t\big) \to 0$ almost surely.

By (B.12), we have

$$\frac{t}{t^{1/\alpha}}\overline{\boldsymbol{\delta}}_t^1 = \frac{1}{t^{1/\alpha}}\mathbf{F}_t\boldsymbol{\delta}_0 - \frac{1}{t^{1/\alpha}}\sum_{j=1}^{t-1}\big(\mathbf{A}^{-1} + \mathbf{W}_j^t\big)\boldsymbol{\xi}_j, \tag{B.17}$$

where the matrices $\mathbf{F}_t$ and $\mathbf{W}_j^t$ are defined back in (B.13) and (B.14). For the non-linear process (2.1), it can be viewed *as if it is a linear process with the $j$-th noise term being $\boldsymbol{\xi}_j + R(\boldsymbol{x}_{j-1}) - \mathbf{A}\boldsymbol{\delta}_{j-1}$.* Hence by (B.12), we have

$$\frac{t}{t^{1/\alpha}}\overline{\boldsymbol{\delta}}_t = \frac{1}{t^{1/\alpha}}\mathbf{F}_t\boldsymbol{\delta}_0 - \frac{1}{t^{1/\alpha}}\sum_{j=1}^{t-1}\big(\mathbf{A}^{-1} + \mathbf{W}_j^t\big)\big(\boldsymbol{\xi}_j + R(\boldsymbol{x}_{j-1}) - \mathbf{A}\boldsymbol{\delta}_{j-1}\big). \tag{B.18}$$

Combining (B.17) and (B.18) yields the difference (cf. Part 4 of Polyak and Juditsky [1992, Proof of Theorem 2])

$$\frac{t}{t^{1/\alpha}}\big(\overline{\boldsymbol{\delta}}_t^1 - \overline{\boldsymbol{\delta}}_t\big) = \frac{1}{t^{1/\alpha}}\sum_{j=1}^{t-1}\big(\mathbf{A}^{-1} + \mathbf{W}_j^t\big)\big(R(\boldsymbol{x}_{j-1}) - \mathbf{A}\boldsymbol{\delta}_{j-1}\big). \tag{B.19}$$

We also recall the assumption that $|R(\boldsymbol{x}_j) - \mathbf{A}\boldsymbol{\delta}_j| \leqslant K|\boldsymbol{\delta}_j|^q$. Hence, it suffices to show the following term vanishes almost surely as $t \to \infty$:

$$J_t = \frac{1}{t^{1/\alpha}}\sum_{j=1}^{t-1}|\boldsymbol{\delta}_j|^q.$$

To show this, first by our calculation of the rate of convergence in Corollary 4,

$$\mathbb{E}\Bigg[\sum_{j=1}^{t-1}\frac{1}{j^{1/\alpha}}|\boldsymbol{\delta}_j|^q\Bigg] = \sum_{j=1}^{t-1}\tilde{\mathcal{O}}\Big(j^{-\rho q\frac{\alpha-1}{\alpha} - \frac{1}{\alpha}}\Big) = \mathcal{O}(1).$$

The last equality holds since $-\rho q\frac{\alpha-1}{\alpha} - \frac{1}{\alpha} < -1$. Hence, we have

$$\mathbb{P}\Bigg[\sum_{j=1}^{t-1}\frac{1}{j^{1/\alpha}}|\boldsymbol{\delta}_j|^q < \infty\Bigg] = 1. \tag{B.20}$$

By Kronecker's lemma, (B.20) implies that $\mathbb{P}[\lim_{t\to\infty} J_t = 0] = 1$. This further implies that the left hand side of (B.19), $t^{1-1/\alpha}\left(\overline{\boldsymbol{\delta}}_t^1 - \overline{\boldsymbol{\delta}}_t\right)$, converges to 0 almost surely.

**Part II.** It remains to show that $t^{1-1/\alpha}\overline{\boldsymbol{\delta}}_t^1$ converges weakly to an $\alpha$-stable distribution.

Define $\overline{\boldsymbol{x}}_t^1 = (\boldsymbol{x}_0^1 + \ldots + \boldsymbol{x}_{t-1}^1)/t$. Since $t^{1-1/\alpha}\left(\overline{\boldsymbol{x}}_t^1 - \overline{\boldsymbol{x}}_t\right) = t^{1-1/\alpha}\left(\overline{\boldsymbol{\delta}}_t^1 - \overline{\boldsymbol{\delta}}_t\right) \to 0$ almost surely, it follows *a fortiori* that $\overline{\boldsymbol{x}}_t^1 - \overline{\boldsymbol{x}}_t \to 0$ almost surely. Hence $\boldsymbol{x}_t^1 - \boldsymbol{x}_t \to 0$ almost surely, due to the well-known theorem that a real-valued sequence converges to zero if and only if the average sequence converges to zero.

Therefore, for the noise decomposition $\boldsymbol{\xi}_{t+1}(\boldsymbol{x}_t) = \boldsymbol{\zeta}_{t+1} + \boldsymbol{m}_{t+1}(\boldsymbol{x}_t)$, the state-dependent component $\boldsymbol{m}_{t+1}(\boldsymbol{x}_t)$ satisfies not only (3.3), i.e.,

$$\mathbb{E}\left[|\boldsymbol{m}_{t+1}(\boldsymbol{x}_t)|^2 \mid \mathcal{F}_t\right] \leqslant K\left(1 + |\boldsymbol{x}_t|^2\right),$$

but also

$$\mathbb{E}\left[|\boldsymbol{m}_{t+1}(\boldsymbol{x}_t)|^2 \mid \mathcal{F}_t\right] \leqslant C\left(1 + |\boldsymbol{x}_t^1|^2\right).$$

Hence, combining the discussion above and Lemma 18, we know that the linear recursion (B.16) defines a process that satisfies Theorem 5. (The only difference is that $\kappa$, instead of (B.15), can be taken from the range $(\rho, 1)$ under the assumption of the current theorem, since by Theorem 3, $\sup_{t\in\mathbb{N}^+} \mathbb{E}[|\boldsymbol{\xi}_t|^p] < \infty$ for every $1 \leqslant p < \alpha$.) It then follows from Theorem 5 that $t^{1-1/\alpha}\overline{\boldsymbol{\delta}}_t^1$ converges weakly to an $\alpha$-stable distribution.

The proof is complete. $\qquad\square$

# C  Additional Technical Background

## C.1  Properties of $\alpha$-Stable Distributions

An $\alpha$-stable distributed random variable $X$ is denoted by $X \sim \mathcal{S}_\alpha(\sigma, \theta, \mu)$, where $\alpha \in (0, 2]$ is the *tail-index*, $\theta \in [-1, 1]$ is the *skewness* parameter, $\sigma \geqslant 0$ is the *scale* parameter, and $\mu \in \mathbb{R}$ is called the *location* parameter. An $\alpha$-stable random variable $X$ is uniquely characterized by its characteristic function: $\mathbb{E}[\exp(iuX)] = e^{-\sigma^\alpha|u|^\alpha(1-i\theta\mathrm{sgn}(u)\tan(\frac{\pi\alpha}{2}))+i\mu u}$, if $\alpha \neq 1$ and $\mathbb{E}[\exp(iuX)] = e^{-\sigma|u|(1+i\theta\frac{2}{\pi}\mathrm{sgn}(u)\log|u|)+i\mu u}$, if $\alpha = 1$, for any $u \in \mathbb{R}$. The mean of $X$ coincides with $\mu$ if $\alpha > 1$, and otherwise the mean of $X$ is undefined. The skewness parameter $\theta$ is a measure of asymmetry. We say that $X$ follows a *symmetric* $\alpha$-stable distribution denoted as $\mathcal{S}\alpha\mathcal{S}(\sigma) = \mathcal{S}_\alpha(\sigma, 0, 0)$ if $\theta = 0$ (and $\mu = 0$). The tail-index parameter $\alpha \in (0, 2]$ determines the tail thickness of the distribution, and $\sigma > 0$ measures the spread of $X$ around its mode. When $\alpha < 2$, $\alpha$-stable distributions have heavy tails so that their moments are finite only up to the order $\alpha$. More precisely, let $X \sim \mathcal{S}_\alpha(\sigma, \theta, \mu)$ with $0 < \alpha < 2$. Then $\mathbb{E}[|X|^p] < \infty$ for any $0 < p < \alpha$ and $\mathbb{E}[|X|^p] = \infty$ for any $p \geqslant \alpha$, which implies infinite variance (see e.g. [Samorodnitsky and Taqqu, 1994, Property 1.2.16]). When $0 < \alpha < 2$, the left tail and right tail of $X$ are described by the formulas:

$$\lim_{x\to\infty} x^\alpha \mathbb{P}(X > x) = \frac{1+\theta}{2}C_\alpha\sigma^\alpha, \qquad \lim_{x\to\infty} x^\alpha \mathbb{P}(X < -x) = \frac{1-\theta}{2}C_\alpha\sigma^\alpha,$$

where $C_\alpha := (1 - \alpha)/(\Gamma(2 - \alpha)\cos(\pi\alpha/2))$ if $\alpha \neq 1$ and $C_\alpha := 2/\pi$ if $\alpha = 1$, (see e.g. [Samorodnitsky and Taqqu, 1994, Property 1.2.15]). The family of $\alpha$-stable distributions include normal, Lévy and Cauchy distributions as special cases, and can be used to model many complex stochastic phenomena [Sarafrazi and Yazdi, 2019, Fiche et al., 2013, Farsad et al., 2015].

## C.2  Domains of Attraction of Stable Distributions

Let $X_i$ be an i.i.d. sequence with a common distribution whose distribution function is denoted as $F$, and let $S_n := X_1 + X_2 + \cdots + X_n$. Suppose that for some normalizing constants $a_n > 0$ and $b_n$, the sequence $S_n/a_n - b_n$ has a non-degenerate limit distribution with distribution function $G$, i.e.

$$\lim_{n\to\infty} \mathbb{P}(S_n/a_n - b_n \leqslant x) = G(x), \tag{C.1}$$

for all continuity points $x$ of $G$, then such limit distributions $G$ are stable distributions and the set of distribution functions $F$ such that $S_n/a_n - b_n$ converges to a particular stable distribution is called its *domain of attraction*.

Next, let us provide a sufficient and necessary condition for being in the domain of attraction of a stable distribution. The class of distribution functions $F$ for which $S_n/a_n - b_n$ converges to $\mathcal{S}\alpha\mathcal{S}(\sigma)$ is called the $\alpha$-stable domain of attraction, and we denote it as $F \in D_\alpha$. Before we proceed, let us recall that a positive measurable function $f$ is *regularly varying* if there exists a constant $\gamma \in \mathbb{R}$ such that $\lim_{t\to\infty} \frac{f(tx)}{f(t)} = x^\gamma$, for every $x > 0$. In this case, we denote $f \in RV_\gamma$, and we say a function $f$ is slowly varying if $f \in RV_0$.

Define the characteristic functions $\phi(u) := \int_{-\infty}^\infty e^{iux} dF(x)$ and $\psi(u) := \int_{-\infty}^\infty e^{iux} dG(x)$, and also define $\lambda(u) := \phi(1/u)$ and $g(u) := \psi(1/u)$ for $u \in [-\infty, \infty] \setminus \{0\}$. We also denote $U(x) := \mathrm{Re}\lambda(x)$ and $V(x) := \mathrm{Im}\lambda(x)$. By Lévy's continuity theorem for characteristic functions (see e.g. Feller [1971, Chapter XV.3]), the convergence in (C.1) is equivalent to $\lim_{n\to\infty} \exp(-ib_n/u)\lambda^n(a_n u) = g(u)$, $u \in [-\infty, \infty] \setminus \{0\}$ uniformly on neighborhoods of $\pm\infty$. Based on this, one can show that (see e.g. ) if (C.1) holds, then $|g(u)|^2 = \exp(-c|u|^{-\alpha})$ for some $\alpha \in (0, 2]$ and $c > 0$ and moreover $-\log|\lambda| \in RV_{-\alpha}$, i.e. $-\log|\lambda|$ is regularly varying with index $-\alpha$. Next, we state a sufficient and necessary condition for being in the $\alpha$-stable domain of attraction.

**Theorem 19** (Geluk and de Hann [2000], Theorem 1). *Suppose $0 < \alpha < 2$. Every $\alpha$-stable random variable $X$ has a characteristic function of the form:*

$$\mathbb{E}[\exp(iuX)] = \exp\left(-\left\{|u|^\alpha + iu(2p-1)\{(1-\alpha)\tan(\alpha\pi/2)\}\frac{|u|^{\alpha-1}-1}{\alpha-1}\right\}\right),$$

*for some $0 \leqslant p \leqslant 1$ with $(1-\alpha)\tan(\pi/2)$ defined to be $2/\pi$ at $\alpha = 1$. The following statements are equivalent:*

*(i) $F \in D_\alpha$.*

*(ii) $1 - F(x) + F(-x) \in RV_{-\alpha}$ and there exists a constant $p \in [0, 1]$ such that*

$$\lim_{x\to\infty} \frac{1 - F(x)}{1 - F(x) + F(-x)} = p.$$

*(iii) $1 - U(x) \in RV_{-\alpha}$ and there exists a constant $p \in [0, 1]$ such that*

$$\lim_{x\to\infty} \frac{xuV(xu) - xV(x)}{x(1 - U(x))} = (2p-1)(1-\alpha)\tan\left(\frac{\alpha\pi}{2}\right)\frac{|u|^{1-\alpha}-1}{1-\alpha}, \qquad u \in \mathbb{R}\setminus\{0\}.$$

Furthermore, [Geluk and de Hann, 2000, Theorem 1] showed that if any of (i), (ii), (iii) holds, then $\lim_{x\to\infty} \frac{1-U(x)}{1-F(x)+F(-x)} = \Gamma(1-\alpha)\cos(\alpha\pi/2)$ and $\lim_{x\to\infty} \frac{V(x)-x^{-1}\int_0^x(1-F(y)-F(-y))\mathrm{d}y}{1-F(x)+F(-x)} = (2p-1)\left(\Gamma(1-\alpha)\sin(\alpha\pi/2) - \frac{1}{1-\alpha}\right)$.

Let us illustrate [Geluk and de Hann, 2000, Theorem 1] with an example of Pareto distribution, which is a power-law distribution widely applied in various fields. A random variable $X$ is said to follow a Pareto distribution (of type I) if there exists some $c > 0$ such that $\mathbb{P}(X > x) = (x/c)^{-\alpha}$ for any $x \geqslant c$ and $\mathbb{P}(X > x) = 1$ for any $x < c$. In this case, $F(x) = 1 - (x/c)^{-\alpha}$ for any $x \geqslant c$ and $F(x) = 0$ for any $x < c$. It follows that $1 - F(x) + F(-x) \in RV_{-\alpha}$ and $\lim_{x\to\infty} \frac{1-F(x)}{1-F(x)+F(-x)} = 1$. Therefore, $F \in D_\alpha$ and the Pareto distribution is in the $\alpha$-stable domain of attraction.

When the tail-index $\alpha \in (0, 2)$, the logarithm of the characteristic function (i.e. $\log \mathbb{E}[e^{iuX}]$) of an $\alpha$-stable random variable $X$ is of the form (see [Gnedenko and Kolmogorov, 1954, equation (12), page 168]):

$$i\gamma u + c_1 \int_{-\infty}^0 \left[e^{iux} - 1 - \frac{iux}{1+x^2}\right]\frac{\mathrm{d}x}{|x|^{1+\alpha}} + c_2 \int_0^\infty \left[e^{iux} - 1 - \frac{iux}{1+x^2}\right]\frac{\mathrm{d}x}{x^{1+\alpha}}, \tag{C.2}$$

where $c_1, c_2 \geqslant 0$ and $\gamma \in \mathbb{R}$. Since the characteristic function uniquely characterizes a probability distribution, the triplet $(c_1, c_2, \alpha)$ uniquely determines an $\alpha$-stable law up to a constant shift $\gamma \in \mathbb{R}$ when $0 < \alpha < 2$. [Gnedenko and Kolmogorov, 1954, Theorem 2, page 175] provides another

sufficient and necessary condition for being in the domain of attraction of an $\alpha$-stable distribution, which complements [Geluk and de Hann, 2000, Theorem 1]. Suppose $0 < \alpha < 2$. Then, the distribution function $F(x)$ belongs to the domain of attraction of an $\alpha$-stable distribution if and only if the following conditions hold: (i) $\lim_{x\to\infty} \frac{F(-x)}{1-F(x)} = \frac{c_1}{c_2}$. (ii) For every constant $\kappa > 0$, $\lim_{x\to\infty} \frac{1-F(x)+F(-x)}{1-F(\kappa x)+F(-\kappa x)} = \kappa^\alpha$. In the case of a Pareto distribution (of type I), for some $c > 0$, we have $F(x) = 1 - (x/c)^{-\alpha}$ for any $x \geqslant c$ and $F(x) = 0$ for any $x < c$. Then we can check that $\lim_{x\to\infty} \frac{F(-x)}{1-F(x)} = 0$ and for every constant $\kappa > 0$, $\lim_{x\to\infty} \frac{1-F(x)+F(-x)}{1-F(\kappa x)+F(-\kappa x)} = \lim_{x\to\infty} \frac{(x/c)^{-\alpha}}{(\kappa x/c)^{-\alpha}} = \kappa^\alpha$. Thus, the Pareto distribution belongs to the domain of attraction of an $\alpha$-stable distribution.

Finally, let us provide a sufficient and necessary condition for being in the domain of normal attraction of a stable distribution.

**Theorem 20** (Gnedenko and Kolmogorov [1954], Theorem 5, page 181). *Suppose $0 < \alpha < 2$. The distribution function $F(x)$ belongs to the domain of attraction of an $\alpha$-stable distribution characterized by* (C.2) *if and only if*

$$F(x) = (c_1 a^\alpha + \alpha_1(x)) \frac{1}{|x|^\alpha}, \qquad \text{for } x < 0, \tag{C.3}$$

$$F(x) = 1 - (c_2 a^\alpha + \alpha_2(x)) \frac{1}{x^\alpha}, \qquad \text{for } x > 0, \tag{C.4}$$

*where $a > 0$ is a positive constant and $\alpha_1(x), \alpha_2(x)$ are functions satisfying $\lim_{x\to-\infty} \alpha_1(x) = \lim_{x\to\infty} \alpha_2(x) = 0$. Indeed, the constant $a$ in* (2.2), (C.3) *and* (C.4) *is the same.*

In the case of a Pareto distribution (of type I), for some $c > 0$, we have $F(x) = 1 - (x/c)^{-\alpha}$ for any $x \geqslant c$ and $F(x) = 0$ for any $x < c$. Then we can check that (C.3) and (C.4) hold with $c_1 = 0$, $\alpha_1(x) \equiv 0$, $c_2 = 1$, $\alpha_2(x) \equiv 0$ and $a = c$. Thus, the Pareto distribution belongs to the domain of normal attraction of an $\alpha$-stable distribution.