# OpenReview forum: "Convergence Rates of Stochastic Gradient Descent under Infinite Noise Variance"
_NeurIPS.cc/2021/Conference — NeurIPS 2021 Poster_

### Official Review · Reviewer_b6Pg · 2021-07-07

**Rating:** 6
**Confidence:** 3

**Summary:**

Previous theoretical analysis of SGD algorithms for convex functions have in general assumed the finitness of the second moment of the stochastic gradient.  The current paper provides convergence rates and generalized central limit theorems under alternative assumptions on the stochastic gradient and the hessian matrices.

**Ethical Concerns:**

I am not aware of ethical concerns regarding the paper.

**Limitations And Societal Impact:**

I am not aware of negative societal impact of their work.

**Main Review:**

The paper defines the concept of $p$-positive definitness for square symmetric matrices, for $p\ge1$. This concept generalizes the concept of positive definite matrices, and the two concepts are the same when $p=2$. Assumption 1 says that the Hessian matrices are uniformly $p$-positive definite. Assumption A2 says that the gradient noise is the sum of two components, the second of which has a finite $p$ moment. Theorem 3 shows convergence results for the SGD algorithm under Assumptions A1 and A2 for a suitable step-size. Under certain assumptions, Theorems 5 and 6 prove central limit theorems for a normalized average of iterates. The paper uses SGD without any modification.

Theorem 3 in the paper is in the same spirit as Theorem 4 of Zhang et al Neurips 2020. While Theorem 3 in the paper requires $p$-positive definitness assumptions on the Hessians and certain assumptions on the stochastic gradients, Theorem 4 in Zhang et al requires a strong-convexity assumption and the unifom finiteness of moments of the stochastic gradients.

The results in the paper are interesting and new to me. On the negative side, the paper contains no numerical experiments, and it is not clear whether the $p$-positive definitness assumption holds in practical applications. Below are more detailed comments or questions.

--In Assumption A2, is $\\{m_t\\}$ a martingale difference sequence with respect to $\\{\cal F_t\\}$?

--The definition of the domain of normal attraction in (4.2) does not coincide with the definition in (2.2).

--line 273: why does Assumption A3 imply that $E[|\xi_t|^p]< \infty$?

--line 340: The calculation of $E[|m_t|^2|x_{t-1}]$ involves  $E[||z_t||^2z_tz_t^T]$. So $E[|m_t|^2|x_{t-1}]$ is not always finite if the second moment of $z$ is finite.

--line 358: for logistic regressions, $y_t$ is binary in practice, and has finite second moment.

--line 112 : typo in "the only a few"


----After feedback, I increased my score----

**Time Spent Reviewing:**

8

---

> ### Author Response · Authors · 2021-08-10
> **Response to reviewer b6Pg**
>
>
> We thank the reviewer for their valuable feedback. We believe we addressed all the concerns. We hope the reviewer could reconsider their overall score.
>
> - **Comparison to [Zhang et. al. 2020]**: We would like to highlight that
> 1. Although both papers are under the infinite noise variance setup, the algorithms that are studied are different. Our work focuses on **vanilla SGD**, whereas [Zhang et. al. 2020] considers **SGD with clipped gradients**.
> 2. As mentioned by the reviewer, in [Zhang et. al. 2020], the noise is assumed to have uniformly bounded $p$-th moment, which is not satisfied even in the linear regression setting. Whereas we assume that the noise is state dependent, covering a wider range of noise settings arising in practice (including linear regression).
> 3. As mentioned by the reviewer, [Zhang et. al. 2020] assumes strong convexity but uses gradient clipping to deal with heavy-tailed noise, whereas we study vanilla SGD and use $p$-PD condition to deal with heavy tailed noise. We emphasize that $p$-PD condition is easily satisfied for a large class of data generating procedures as argued below.
>
>
> - **No experiments**: We see our contributions as mainly theoretical, but if the paper is accepted we will conduct experiments to demonstrate i) the $L^p$ convergence of vanilla SGD in the infinite noise variance setting ii) the $\alpha$-stable limit of the Polyak-Ruppert averaging for the examples discussed in Section 5. However, we must mention that purely theoretical papers without any empirical results are not that uncommon at NeurIPS and ICML conferences, to name just a few:
>
> [1] Sato, Issei, and Hiroshi Nakagawa. "Approximation analysis of stochastic gradient Langevin dynamics by using Fokker-Planck equation and Ito process." ICML, 2014.
>
> [2] Xu, Pan, et al. "Global convergence of Langevin dynamics based algorithms for nonconvex optimization. NeurIPS’18. (spotlight)
>
> [3] Nguyen, Than Huy, Umut Simsekli, and Gaël Richard. "Non-asymptotic analysis of Fractional Langevin Monte Carlo for non-convex optimization." ICML, 2019.
>
> - **$p$-PD does not hold in general**: We note that 1-PD (diagonal dominance) implies $p$-PD for any $p\geq2$ as proved in Section A.2 and illustrated in Figure 1. Even in this extreme case (this is the most restrictive this assumption can get), it is easy to verify $p$-PD condition in, for example, optimization problems defined by linear regression and GLMs. As an example, consider a data generating process where inputs are multivariate Gaussian, i.e. $z_t\sim N(0,I)$, which is commonly used in statistics and termed as random Gaussian design [1,2,4] (intimately tied to *data whitening*). In linear regression, as stated in l342, we have $\nabla^2 f(x) =I$ which is always 1-PD, thus $p$-PD for any $p\geq 1$. This argument can also be made for GLM minimization problems as discussed in Section 5.2. It should be noted that, just like strong convexity can be violated, $p$-PD may also be violated in certain optimization problems. Understanding the the behavior of SGD in such scenarios (together with infinite noise variance) is a fruitful ground for future work.
>
>     Finally, we emphasize that the diagonal dominance property is a very important subject in numerical linear algebra, and has very interesting properties and consequences [5]. To our knowledge, our work is to first to provide an interpolation between this class of matrices and another well-studied class of matrices, the positive semi-definite cone. We believe that there are interesting properties of $p$-PD matrices to be investigated, which would be another fertile ground for future research.
>
> - **Martingale difference**: The reviewer is correct. $( m_t)_t$ is a martingale difference sequence adapted to the filtration $(\mathcal{F}_t )_t$.
>
> - **Domain of normal attraction**: We thank the reviewer for bringing this inconsistency to our attention. (4.2) is a special case of (2.2). We will make these two statements consistent with each other in the revised version.
>
> - **l273**: This is a classical result in the literature of stable distributions. See e.g. [3, Corollary 16.30].
>
> - **l340**: We thank the reviewer for bringing this to our attention. Indeed, we need finite fourth moment of the covariates in this example, similar to the GLM example.
>
> - **l358**: The reviewer is correct in that a response generated by the logistic regression **model** cannot be heavy-tailed. However, we see (5.4) only as a minimization problem, without assuming a particular statistical model. In other words, as stated in line 356, model can be misspecified. We will clarify this further in the revised version.
>
> - **l112**: Thank you for noticing this typo. We will correct this typo in a revised version of our work.
>
> We would be happy to clarify any further concerns/questions in the discussion period.
>
> ---
> [1] Lucien Birgé. "Model selection for Gaussian regression with random design." Bernoulli 2004
>
> [2] Mohsen Bayati, Andrea Montanari. "The LASSO risk for Gaussian matrices", Transaction on Information Theory, 2011
>
> [3] Klenke, Achim. "Probability theory: a comprehensive course". Springer Science and Business Media, 2013.
>
> [4] Hua Wang, Yachong Yang, Zhiqi Bu, Weijie J. Su "The Complete Lasso Tradeoff Diagram", Advances in Neural Processing Systems, 2020
>
> [5] Gene H. Golub, and Charles F. Van Loan. "Matrix computations. Johns Hopkins studies in the mathematical sciences." 1996.

---

### Official Review · Reviewer_vEYC · 2021-07-17

**Rating:** 6
**Confidence:** 3

**Summary:**

The paper considers the convergence theory of SGD under assumptions that does not imply finite variance of the noise of the stochastic gradient for a class of strongly convex functions. A convergence rate is proven for an existing moment of the iterates of SGD and a CLT is proven for the averaged of the iterates (towards an alpha stable distribution).


**Limitations And Societal Impact:**

Some limitations are discussed. Another limitation is that this work relies on a lot of modelization/assumptions about the noise. Even if it is proven that this work covers OLS and GLM, I am not sure that it can go beyond.

**Main Review:**

Originality: This work studies the convergence of SGD in the case where the noise has infinite second moment. While this covers interesting practical cases, this is in contrast with most of the works on the convergence of SGD.

Quality: The convergence results are discussed in details and the paper is well written, although technical. A good literature review is provided for stochastic approximation under assumptions that do not imply finite variance of the noise. Besides, a CLT is provided for the average of the iterates, towards an alpha stable distribution. My main concern is wrt to the assumptions made in theorems. First, the work is limited to strongly convex functions and second the assumptions made on the noise do not seem easy to check. I must admit that a real effort is made to prove that the assumptions are satisfied for Ordinary Least Square and Generalized Linear Models, but this still relies on some hypotheses.

Clarity: The work is technical and more effort should be made on clarifying the paper. The intro and literature review are quite technical. The choice of alpha stable distribution in Assumption 3 is not really justified. It seems specific. Why this choice? Do we know the parameters of the alpha stable distribution? Specific questions:
- Th 5: Alpha is given by assumption 3, am I correct? Do we know alpha easily?
- l.290: "Indices"?
- Could you explain how to obtain 5.3 and l.363 ?

Significance: The work considers the infinite variance noise, which an important setting is practice. However it still requires some assumptions, so the work is not that general.

**Time Spent Reviewing:**

3

---

> ### Author Response · Authors · 2021-08-10
> **Response to reviewer vEYC**
>
> We thank the reviewer for their valuable feedback.
>
> - **Limited to strong convexity**: We agree that extending the scope of our work is an interesting and important direction. However, we emphasize that our work is to first to establish the convergence rates of SGD in the infinite noise variance setting in such generality. We also highlight that the limit distribution of the Polyak-Ruppert averaging of SGD iterates is in general established under a form of strong convexity (see e.g. [1-5]), and our paper is the first to show this in the infinite noise variance setting.
>
> [1] Boris T Polyak and Anatoli B Juditsky. Acceleration of stochastic approximation by averaging. SIAM Journal on Control and Optimization, 1992
>
> [2] Panos Toulis and Edoardo M Airoldi. Asymptotic and finite-sample properties of estimators based on stochastic gradients. The Annals of Statistics, 2017
>
> [3] Weijie Su and Yuancheng Zhu. Statistical inference for online learning and stochastic approxi- mation via hierarchical incremental gradient descent, arXiv, 2019
>
> [4] Nilesh Tripuraneni, Nicolas Flammarion, Francis Bach, and Michael I Jordan. Averaging stochastic gradient descent on riemannian manifolds. Conference on Learning Theory, 2018
>
> [5] Yixin Fang, Jinfeng Xu, and Lei Yang. Online bootstrap confidence intervals for the stochastic gradient descent estimator. The Journal of Machine Learning Research, 2018
>
> - **Noise assumptions**: We agree with the reviewer that Assumption 2 on the noise is more involved compared to recent works on SGD with heavy-tailed noise.
> This is due to the fact that our noise model in Assumption 2 is significantly more general. Consider for instance, the noise model in the
> recent work Zhang et al. [6] which assumes $\mathbb{E}[\|\xi_{t}(x)\|^p] \leq \sigma^p$ for all $x$, where $\sigma$ does not depend on $x$. Therefore, this noise model cannot handle state-dependent noise, and it is easy to verify that even linear regression does not satisfy this noise model. In contrast, in Section 5.1, we explicitly verify that our noise model is satisfied in the linear regression setting.
>
> [6] Zhang et al.  Why are adaptive methods good for attention models? In Advances in Neural Information Processing Systems, 2020.
>
> - **Clarity**: Taking the reviewer's advice, we will improve the presentation by making the introduction and the literature review less technical.
>
> - **$\alpha$-stable limit in Assumption 3**:
> 1. Stable distribution emerges as the limit distribution in the Generalized Central Limit Theorem (GCLT) due to Gnedenko and Kolmogorov [7], where the case $\alpha =2$ corresponding to the classical Central Limit Theorem (CLT) with Gaussian limit. Therefore, $\alpha$-stable distribution should be seen as a generalization of Gaussian distribution, emerging as the limit distribution of average of i.i.d. random variables. Hence, it is not a choice we made in Assumption 3, it naturally appears in our theory as it is the limiting distribution in GCLT.
>
>     To demonstrate what Assumption 3 imposes on the vectors $\zeta_t$, let's consider the special (and perhaps not very interesting) case $\alpha=2$. In this case, (4.3) in Assumption 3 roughly means *... for i.i.d. random vectors $\zeta_t$ that satisfy a classical CLT ...*. Therefore, the condition holds for any distribution that admits a classical CLT. For example, random variables with finite variance falls into this category (Lindeberg-Levy CLT); however, the condition as stated is more general.
>
>     Similarly, we remark that the assumption *"in the domain of normal attraction"* actually encompasses the largest class of noise distributions that produce the $\alpha$-stability result which is a generalization of the classical CLT. Replacing *"in the domain of normal attraction"* with *"having power law tail"*, for example, does not invalidate the theorems but makes the conditions more restrictive.
>
> 2. The $\alpha$-stable distribution in the Assumption 3 is symmetric in accordance with the results of Gnedenko and Kolmogorov [7]. This simply means that the limit law $\mu$ is equal to $\mathcal{S}_\alpha(\sigma,0,0)$ where the distribution $\mathcal{S}_\alpha$ is defined in line 144 of the main text with an explicit characteristic function. In Section C in the supplementary document, we provided a detailed discussion on $\alpha$-stable limits complimenting the brief introduction given in the main text (lines 135-161).
>
> 3. The reviewer is correct that $\alpha$ is given by Assumption 3.
>
> 4. The parameter $\alpha$ is easily estimable. For instance, in the linear regression example we discussed in Section 5.1, $\alpha$ is the tail index of the response $y_t$ for which there are well-established estimators in the literature. Therefore by simply estimating the tail index of the response variables, one can obtain a  good estimate for $\alpha$ [8].
>
> [7] B. V. Gnedenko and A. Kolmogorov. "Limit Distributions for Sums of Independent Random Variables". Addison Wesley, Cambridge, MA, 1954
>
> [8] Mohammad Mohammadi, Adel Mohammadpour, and Hiroaki Ogata. "On estimating the tail index and the spectral measure of multivariate $\alpha$-stable distributions." Metrika 78.5 (2015): 549-561.
>
>
> - **Steps leading to (5.3) and l.363**:
> 1. For the linear regression example, we have the **full gradient** $\nabla f(x)=\mathbb{E}[zz^\top] x - \mathbb{E}[zy]$ as given in (5.1) and the SGD update is given as
> $$ x_t = x_{t-1} - \gamma_t (z_t z_t^\top x_{t-1} - z_t y_t) = x_{t-1} - \gamma_t (\nabla f(x_{t-1}) + \xi_t).$$
> Therefore, we can find the noise via $\xi_t =z_t z_t^\top x_{t-1} - z_ty_t- \nabla f(x_{t-1})$, which yields
> $$\xi_t = (z_t z_t^\top x_{t-1} - z_ty_t) - (\mathbb{E}[zz^\top] x_{t-1} - \mathbb{E}[zy]).$$
> Rearranging and defining $\zeta_t = E[zy]-z_t y_t$ which is iid and heavy tailed; and $m_t = (z_t z_t^\top - E[z z^\top])x_{t-1} $ which is the state dependent component, we obtain (5.3).
>
> 2. For the GLM example, we similarly have the **full gradient** $\nabla f(x)=\mathbb{E}[z\psi'(z^\top x)] - \mathbb{E}[zy] + \lambda x$ as given in (5.5) and the SGD update reduces to
> $$ x_t = x_{t-1} - \gamma_t (z_t \psi'(z_t^\top x_{t-1}) - z_t y_t + \lambda x_{t-1}) = x_{t-1} - \gamma_t (\nabla f(x_{t-1}) + \xi_t).$$
> Therefore, we can find the noise via $\xi_t =z_t \psi'(z_t^\top x_{t-1}) - z_t y_t + \lambda x_{t-1}- \nabla f(x_{t-1})$ which yields
> $$\xi_t = (z_t \psi'(z_t^\top x_{t-1}) - z_ty_t+ \lambda x_{t-1}) - (\mathbb{E}[z\psi'(z^\top x_{t-1})] - \mathbb{E}[zy] + \lambda x_{t-1}).$$
> Rearranging and defining $\zeta_t = E[zy]-z_t y_t$ which is iid and heavy tailed; and $m_t = (z_t \psi'(z_t^\top x_{t-1}) - E[z \psi'(z^\top x_{t-1})]) $ which is the state dependent component, we obtain l363.
>
> - **Minor comments**: Reviewer's minor comments will be addressed carefully.
>
> We would be happy to clarify any further concerns/questions in the discussion period.

---

> > ### Comment · Reviewer_vEYC · 2021-08-17
> > **Thanks**
> >
> > I am satisfied with this answers.
> >
> > The assumptions made are quite involved but I agree that this is inherent to this kind of works that go beyond traditional assumptions.

---

### Official Review · Reviewer_DYic · 2021-07-18

**Rating:** 6
**Confidence:** 4

**Summary:**

This paper concerns the convergence rate of SGD with heavy-tail noise. In particular, the considered gradient noise consists of a finite-variance part (corresponds to the multiplicative noise in SGD for least square) and an infinite-variance part (corresponds to the additive noise in SGD for least square). The main results can be viewed as a law of large number and a central limit theorem in the absence of finite-variance. First the authors show that when the loss is "strongly convex" and "smooth" in the $p$-PD sense and the heavy-tail part noise has finite $p$-moment, SGD (with decaying stepsize) iterates converges to the correct optimum, and the convergence rate is justified in the p-norm sense. Moreover, if the heavy-tail noise is the domain of normal attraction of some stable distribution, then the properly normalized SGD iterates also converges to the stable distribution. Finally, the heavy-tailed noise model is justified from two linear model examples.

**Limitations And Societal Impact:**

See above.

**Main Review:**

# Pros:
+ The paper is organized well and is particularly clear in its math (as a CS paper).
+ The presented convergence rate and CLT for SGD with heavy-tailed noise are novel in machine learning to my knowledge. I am not sure how novel these are in probability theory.
+ Technically, the introduced p-PD notions could be interesting tools for ML people to know. Again, not sure whether or not these are interesting for probability theorists.


# Cons:
- Several important quantities are hidden in the rate in Thm3, e.g., dependence on $x_1$, condition numbers (as the hessian is assumed to be uniform $p$-PD), dimension factors, the noise $p$-moment bound, etc. This limits the importance of the theorem. The presented rate is interesting assuming the above are all constants, but more discussions on the roles of these quantities could make the result even more interesting & inspiring.
- Could you comment on SGD with constant, large stepsize and with iterate-averaging/tail-averaging? In the finite-variance cases such SGD setup achieves optimal rate & I am wondering how it would be in the infinite-variance cases. Is there essential difference other than switching to a different norm?

# Small issues:
* Eq below l.145. $delta$ could be boldface.
* l.322. coefficients -> coefficient.
* Eqs below l.652. The first inequality is not rigorous as $x^b_t$ depends on $x_t$. I believe this is easily fixable and should not affect the correctness of the final claim. Please confirm.

# Overall:
As an ML person who has limited probability background, I think this paper is clear and clean, and is a good reading for understanding SGD in the heavy-tail noise cases. However I am not sure how novel the results to probability theorists. On the one hand, the two main results, Thm3 and Thms 5/6, seem to be applications of developed large number thm and CLT from existing (and quite old) probability papers (perhaps with a little bit standard optimization tricks) --- I could miss some points here and please correct me if so. Thus I am not sure how probability theorists view these results. On the other hand, the effects of initialization, condition numbers, dimension factors, noise bound are very important for rates for ML problems, but these are all hidden in the presented result. Thus the current version is a bit unsatisfactory. It is very hard for me to make a decision, currently I lean slightly to a weak acceptance.







-----
Post-Rebuttal: I tend to maintain my original evaluation for this paper after discussions with the authors. The contribution is solid in my perspective. But there are also several technical dissatisfactions about the current results (e.g., unable to track important constants like condition number).

**Time Spent Reviewing:**

6

---

> ### Author Response · Authors · 2021-08-10
> **Response to reviewer DYic**
>
> We thank the reviewer for their valuable comments.
>
> - **Several quantities are hidden**: We note that the majority of these quantities can be tracked throughout our proof. However, the main bottleneck is the use of Lemma 9 due to Fabian 1967. This lemma is key to our analysis; however, we are not aware of a version of this lemma that allows us to keep track of all the aforementioned constants.
>
>     That being said, we shall reiterate that our results form the first theoretical evidence that shows that **without any modification**, SGD can still converge to the optimum in $L^p$, even when it does not converge in $L^2$. In this sense, we believe that our results significantly improve over the state of the art in this respect and form the first steps towards deriving more explicit bounds in terms of, e.g., condition numbers and dimension factors.
>
>
> - **Constant step-size SGD**: We emphasize that our framework requires decreasing step-size schedule, and does not cover constant step-size SGD. We provide the following comments regarding constant step-size SGD. We kindly ask the reviewer to reach out to us during the discussion period if there are additional questions/concerns.
>
> 1. We note that constant step-size SGD does not converge to the minimizer in the asymptotic limit as $t \to \infty$ [2]. In other words, constant step size SGD will give a biased result, and the amount of this bias is determined by the local growth behavior of the objective function [2,3]. In certain cases, smaller bias can be obtained by choosing a smaller step-size; however, in the large step-size regime, constant step-size SGD cannot converge to an arbitrary neighborhood of the minimizer in general.
>
> 2. The convergence analysis is carried out by interpreting the SGD iterations as a Markov chain [2,3]. Similar to the decreasing step-size setting, Polyak-Ruppert averaging of the constant step-size SGD also admits a CLT [3]. However, when the step size is large, SGD iterates are known to exhibit heavy-tails due to multiplicative noise [1,4], and $\alpha$-stable limits are established in the simplistic linear regression setting [1].
>
> 3. The optimality of Polyak-Ruppert averaging requires a decreasing step-size [5], and the asymptotic covariance of the constant step-size SGD does not achieve Cramer-Rao Lower Bound (CRLB) in general [3]; thus, it is not asymptotically efficient.
>
> 4. As commented in line 312, it is not even clear what the above optimality translates to in the infinite variance case. There are generalizations of CRLB (and information inequality) for the infinite variance case; however, the analogy is by no means straightforward.
>
> 5. Other than choosing an appropriate norm, one should also ensure contraction in that particular norm. This is where our analysis requires uniform $p$-PD condition. In the case of constant step-size, establishing a $\alpha$-stable limit in the general case may require different techniques since in our current proof, control over various terms are established by making use of the decreasing step-size schedule. Although not covered by our current analysis, we agree with the reviewer that this is a fertile ground for future work.
>
>
> - **Minor comments**: Reviewer's minor comments will be addressed carefully. Indeed, the minor issue in l652 does not affect the correctness of our argument since we rely on the uniform $p$-PD condition. We will update the steps by first using the submultiplicative norm property inside the expectation and next using the uniform $p$-PD condition. We will address this issue in the updated version of our paper.
>
> We would be happy to clarify any further concerns/questions in the discussion period.
>
> ---
>
> [1] M. Gurbuzbalaban, U. Simsekli, L. Zhu. "The heavy-tail phenomenon in sgd", ICML 2021
>
> [2] A. Dieuleveut, A. Durmus, F. Bach. "Bridging the gap between constant step size stochastic gradient descent and markov chains", The Annals of Statistics, 2020
>
> [3] L. Yu, K. Balasubramanian, S. Volgushev and M. Erdogdu, An Analysis of Constant Step Size SGD in the Non-convex Regime: Asymptotic Normality and Bias,  arxiv 2020
>
> [4] L. Hodgkinson and M. W. Mahoney. "Multiplicative noise and heavy tails in stochastic optimization", 2020
>
> [5] B. T. Polyak and A. B. Juditsky. "Acceleration of stochastic approximation by averaging. SIAM Journal on Control and Optimization", 1992

---

> > ### Comment · Reviewer_DYic · 2021-08-13
> > **Need more clarifications**
> >
> > Thank you for the response, but I am confused after reading it:
> >
> > * The response on the hidden quantities confuses me. It is not clear which are the "majority" quantities that can be tracked and what is their order in the obtained bound. Moreover, the authors mentioned that Lemma 9 prevents several quantities to be tracked. The authors should be more specific what are they. I believe the impact of this paper could be affected in this regard.
> >
> > * To be more specific, my second question is about constant-stepsize SGD with *iterate-averaging or tail-averaging*. It is clear that without averaging, constant-stepsize SGD could diverge.
> >
> > * According to the response, it seems the major differences in the analysis between finite-variance SGD vs. heavy-tail SGD are (1) switching to a different $p$-norm, and (2) switching to work with a different $p$-PD matrix space. Is that correct?

---

> > > ### Author Response · Authors · 2021-08-15
> > > **Second response to reviewer DYic**
> > >
> > > We thank the reviewer for their stimulating questions. Please see our point by point response below.
> > >
> > > - **Explicit constants** We agree with the reviewer that the dimension and the condition number dependency of the convergence rate is important. Especially in recent line of work in optimization, rates are often stated with an explicit dependence on these quantities, to explain their impact in high-dimensional and/or ill-conditioned settings. That said, **all of these works** are in the **finite noise variance** setting, which allows them to track constants throughout their analysis.
> > >
> > >     Similarly, as mentioned in our previous response, the dependency on the dimension as well as the condition number can be tracked in our proof until we apply Chung's lemma (Lemma 9). This corresponds to line 669 in the supplement, which is where we lose track of the aforementioned constants. Unfortunately, this Lemma is crucial to obtain a convergence rate and provides a result  without explicit constants. An exact characterization of the constants (that depend on the condition number, dimension etc) in our Theorem 3 requires developing an **explicit version of Chung's lemma** that can be applied to the infinite noise variance setting, which is **currently unavailable** to our knowledge. In a recent preprint, authors were able to prove Chung's lemma with explicit constants in the setting when $\rho=1$ where step-size is $\gamma_t \asymp t^{-\rho}$ (see [1, Lemma 1]).
> > > Unfortunately, the case $\rho=1$ corresponds to a very fast decaying step-size schedule, for which we cannot establish convergence in the infinite variance setting; thus, we cannot use this version of Chung's lemma. Proving Chung's lemma with explicit constants for $\rho <1$ is indeed a very interesting direction; however, it is by no means straightforward and left for another study.
> > >
> > >     In summary, we make the following points:
> > >     1. We acknowledge this limitation of Theorem 3; however, this is due to the state of current machinery applicable in the infinite noise variance setting. We will explicitly state this in our paper, also by highlighting it as an important future direction.
> > >     2. The above limitation is not relevant to our second principal contribution (Theorems 5 and 6), since the $\alpha$-stable limit of the SGD iterate averaging holds for $t \to \infty$; hence, this result will not benefit from an $L^p$ rate that has explicit dimension and condition number dependency (Theorem 3).
> > >
> > >     [1] Kwangjun Ahn and Suvrit Sra, "On Tight Convergence Rates of Without-replacement SGD", arxiv preprint https://arxiv.org/pdf/2004.08657.pdf
> > >
> > > - **Constant step-size SGD** If the step-size $\gamma$ is constant, even with iterate averaging (or tail averaging) the constant step-size SGD will not converge to the optimum $x^*$ in general. This is true for example, for strongly convex objectives -- an exception is quadratic minimization problems which is too restrictive [3]. A comment from [2, pages 2 and 3] (mutatis mutandis):
> > >
> > >     >Let $\bar{x}_t = \frac{x_1 + x_2 + \dots +x_t}{t}$ be the averaged iterate sequence and $\bar{x}_\infty$ denote the limit of $\bar{x}_t$ as $t\to\infty$. The deviation between iterate averaging $\bar{x}_t$ and the global optimum $x^*$ is thus composed of a stochastic
> > > part $\bar{x}_t- \bar{x}_\infty$ and a deterministic part $\bar{x}_\infty - x^*$, which characterizes the bias. For quadratic functions, it turns out that the deterministic part vanishes [3], that is, $\bar{x}_\infty = x^*$ and thus averaged SGD with a constant step-size does converge. However, it is not true for general objective functions where we can only show that $\bar{x}_\infty - x^* = \mathcal{O}(\gamma)$
> > > and this deviation is the reason why constant step-size SGD (even with iterate averaging) is not convergent.
> > >
> > >     Therefore, it can be shown that $\bar{x}_t - x^* = \mathcal{O}(\gamma)$ for large enough $t$, where $\gamma$ is the constant step-size (see e.g. [2, Equation (9)]. In the large step-size setting, constant step-size SGD will have a large bias from the minimizer.
> > >
> > >     [2] Aymeric Dieuleveut. Alain Durmus. Francis Bach. "Bridging the gap between constant step size stochastic gradient descent and Markov chains." Ann. Statist. 48 (3) 1348 - 1382, 2020 https://arxiv.org/pdf/1707.06386.pdf
> > >
> > >     [3] Francis Bach and Eric Moulines. "Non-strongly-convex smooth stochastic approximation with
> > > convergence rate O(1/n)". Advances in Neural Information Processing Systems (NeurIPS), 2013
> > >
> > > - **Technical innovations** At a very high level, the two major differences in the analysis between finite-variance SGD vs. heavy-tail SGD are 1. switching to a different norm, and 2. switching to work with a different $p$-PD matrix space. However, the rest of the analysis doesn't carry on exactly the same as the finite variance setting.
> > >
> > >     We list a few key technical steps in our proof that are not highlighted in the main text.
> > >
> > >     1. **Lemma 7**: This is one of the most important technical contributions, and maybe of independent interest. This result allows us to generalize to martingales in the heavy-tailed setting. It states that the p-th moment for $p < 2$ of a martingale without square-integrability assumption (infinite variance) can also be bounded by the sum of the $p$-th moments of its martingale difference sequence, at the cost of a multiplicative constant. A similar result for **i.i.d. random variables** recently appeared in [4, Lemma 4.2] (as acknowledged by the authors "key to their analysis"); however, our Lemma applies to **martingales** as needed in SGD analysis; thus it is significantly more general. Therefore, this result may be of independent interest as the i.i.d. version appeared in a mean estimation paper.
> > >
> > >     [4] Yeshwanth Cherapanamjeri, Nilesh Tripuraneni, Peter L. Bartlett, Michael I. Jordan. "Optimal Mean Estimation without a Variance", arxiv preprint 2020 https://arxiv.org/pdf/2011.12433.pdf
> > >
> > >     2. Switching to $p$-PD does not directly imply convergence in the respective $L^p$ norm. For this, we established an equivalence between $p$-PD and the contraction in $p$-norm for **any sufficiently small** step size, which is another step that is key to our analysis. These are summarized in Theorems 10 and 11 in the supplement, which we did not highlight in the main text.
> > >
> > >     3. We also proved the **correct** version of an $L^p$ expansion inequality which is summarized in Lemma 8. This result
> > > can be thought of the correction as well as the extension of
> > > Theorem 3 in Krasulina' work, which contains an error in ignoring the signum function sign(x) in this inequality.
> > >
> > >     Finally, the above innovations together with the $L^p$ convergence are used in the proof of Theorems 5 and 6 where we show
> > > the $\alpha$-stable limit for the SGD iterate average. The proof of this work relies on a delicate decomposition of the SGD iterate averaging, but details are skipped in the main text. We emphasize that this result is by no means straightforward given the prior work. We are aware that many important key steps are hidden in the supplement, and we will make our best effort to highlight them in the main text, also by making them accessible to general audience.
> > >
> > > We again thank the reviewer for their stimulating questions, and kindly ask them to follow up on any of the above remarks if we can provide any additional clarification.

---

### Official Review · Reviewer_2Gbm · 2021-07-22

**Rating:** 6
**Confidence:** 3

**Summary:**

The authors analyze SGD for a class of strongly-convex functions and under (possibly) infinite noise variance, i.e., the pth moment of the noise exists for $p \in [1,2)$, but noise variance is not assumed to be bounded. Such noise behavior is defined as heavy-tailed. Additionally, the noise is assumed to be state-dependent such that the noise vector has a component whose variance is bounded by a function of the query point (iterates). For standard analysis of SGD, noise vectors are assumed to be independent of query points and of each other.
Specifically, for a class of strongly convex functions (denoted as p-positive definite (p-PD), p=2 being positive definite Hessian) and under heavy-tailed noise, the authors:

-	Quantify the convergence rate with respect to (w.r.t.) distance to unique optimum in $L^p$, which approaches the optimal rate in strongly convex setting as $p \rightarrow 2$. This result is achieved for SGD without any modifications and under state-dependent noise model.
-	Prove that normalized Polyak-Ruppert average (uniform average) converges weakly to an $\alpha$-stable distribution, under additional assumptions on the noise and the Hessian. Central Limit Theorem (CLT) does not hold under heavy-tailed noise model.

In order to reinforce the setting they consider and verifiability of their assumptions in practice, authors discuss about the real-world examples that admit heavy-tailed noise and satisfy p-PD assumption.


**Limitations And Societal Impact:**

They address some limitations of their results in comparison to prior work. The authors do not mention potential negative societal impact of their work. Given that this is a theory paper, the authors might have thought that it is less likely to have foreseeable negative societal impact.

**Main Review:**

This work tries to extend SGD for heavy-tailed noise for a class of strongly-convex problems satisfying p-PD condition. This condition could be interpreted as a generalization of positive definiteness of Hessian and interpolates between diagonally dominant matrices with non-negative diagonal and positive definite matrices. They also consider a more general noise model where they assume there exists a component of the noise vector that grows with the norm of the query point. In my opinion, these are the main novelties of this paper that builds onto existing work and approaches.

In that perspective, to the best of my knowledge, they adequately cite related work and explain the differences/similarities between their work and the prior ones while emphasizing their improvements.

Authors provide a set of main convergence results as well as limit behavior of the sequence generated b SGD. These results are proven in the supplementary material, however, the implications of these results are provided in the main text through intuitive remarks and technical discussions following the main theorems. Authors also try to verify their setting and validity of the assumptions via particular examples; least squares and generalized linear models.

Overall, the paper is written in a clear way. Problem setting, as well as the assumptions accompanying it are clearly stated and discussed in comparison to existing work. Main contributions are presented to the reader in a separate paragraph at the end of introduction. Main results are organized in a proper way and it is easy to follow the core contributions. Assumption 1 is not defined but referred to several times in the appendix. It is possible to deduce what was meant by Assumption 1 from the context, but it should be defined and referenced clearly.

Relying on the existing results, behavior of stochastic methods under heavy-tailed noise has attracted attention again recently and has been studied in different contexts.

[Zhang et al., 2020] considers heavy-tailed noise without state-dependence. Under $L$-smoothnness, they prove convergence rates for non-convex functions. Under heavy-tailed $\text{\bf stochastic gradients}$ rather than heavy-tailed noise, they show comparable convergence rate for strongly-convex functions. This paper focuses on a class of strongly-convex functions and show convergenve rates under state-dependent heavy-tailed noise. Given the results proposed by [Zhang et al., 2020], this work provides a nice generalization on the heavy-tailed noise model, but its impact is comparatively limited in terms of convergence analysis.

[Simsekli et al., 2019] considers nonconvex problems under heavy-tailed stochastic gradients. With Holder smoothness assumption on the gradient of the objective, they show convergence rates to first-order stationary points. The setting of this paper and [Simsekli et al., 2019] do not overlap and we might say the results are complementary.

As for the limiting behavior of the normalized average iterate, given the results by [Polyak and Juditsky, 1992], Assumption 3 and assumption in Eq. (4.3), Theorem 5 and 6 provide some nice results, but their significance is limited in my opinion.

Please find some of my particular comments below:
- The authors say that Eq. (4.3) is a standard assumption. I do not completely agree with this statement. Authors also state that this condition holds for Hessian Lipschitz objective, but Hessian-Lipschitzness is a rather strong assumption already.
- For proving $\alpha$-stability results, it is assumed that i.i.d. component of the noise is assumed to be in the domain of normal attraction of
an $\alpha$-stable distribution. In my opinion, this makes the results of Theorem 5 and 6 less interesting from a theoretical point of view.


**Time Spent Reviewing:**

6

---

> ### Author Response · Authors · 2021-08-10
> **Response to reviewer 2Gbm**
>
> We thank the reviewer for their valuable comments.
>
>
> - **Comparison to [Zhang et. al. 2020]**: We would like to highlight that
> 1. Although both papers are under the heavy-tailed noise setting, our work focuses on vanilla SGD, whereas [Zhang et. al. 2020] considers **SGD with clipped gradients**. We make a case for vanilla SGD in the heavy-tailed regime that even **without any modifications**, SGD can converge.
> 2. As mentioned by the reviewer, in [Zhang et. al. 2020], the noise is assumed to have uniformly bounded $p$-th moment, which is not satisfied even in the linear regression setting. Whereas we assume that the noise is state dependent, covering a wider range of practical problems.
> 3. As mentioned by the reviewer, [Zhang et. al. 2020] assumes strong convexity but uses gradient clipping to deal with heavy-tailed noise, whereas we study vanilla SGD and assume $p$-PD to deal with heavy tailed noise. We emphasize that $p$-PD is easily satisfied for a large class of data generating procedures as discussed in Section 5 of our paper.
>
> - **Local smoothness**: We note that Equation (4.3) is identical to Assumption 3.2 in Polyak and Juditsky (1992); further, we emphasize that (4.3) needs to hold **only locally**. We are not aware of a distributional convergence result for Polyak-Ruppert averaging of the SGD without making this type of local smoothness assumption on the Hessian.
>
>
> - **Domain of normal attraction**: We remark that the assumption *in the domain of normal attraction* actually encompasses the **largest possible class** of noise distributions that produce the **multivariate** $\alpha$-stability result. Replacing *in the domain of normal attraction* with *having regularly varying tails like power law*, for example, does not invalidate the theorems but makes the conditions more restrictive.
>
>
> - **Minor comments**: Reviewer's minor comments will be addressed carefully. In particular, we will re-state Assumption 1 in the appendix for clarity, which is currently only stated in the main text (line 210).
>
>
> We would be happy to clarify any further concerns/questions in the discussion period.

---

### Decision · Program_Chairs · 2021-09-27

**Decision:**

Accept (Poster)

**Comment:**

This paper examines the convergence of stochastic gradient descent in strongly convex minimization problems. The novelty of the analysis is that the authors do not assume that the variance of the gradient queries is finite; instead, they consider "heavy-tailed" gradient noise models with bounded moments for some $p\in[1,2)$ - but not necessarily for $p=2$ or higher.

This paper received almost universally positive reviews during the review phase, and the only "weak reject" recommendation was changed to a "weak accept" after the authors addressed the reviewer's concerns. As a result, during the committee discussion, a consensus was reached early on to make an "accept" recommendation.